# Contextual Forgetting: Mitigating Knowledge Obsolescence for Safe Lifelong Robot Learning

## Abstract

Lifelong Robot Learning, in its pursuit of general intelligence, confronts a critical yet overlooked challenge: endogenous safety risks arising from "knowledge obsolescence." When a once-optimal policy becomes detrimental after an environmental shift, the conventional Continual Learning (CL) paradigm, which focuses on "remembering," lacks an active "forgetting" mechanism, posing significant risks in the physical world. To address this, we introduce "Contextual Forgetting," a novel mechanism, and design a Knowledge Validity Module (KVM). The core of KVM is a principled risk assessment framework based on an Energy-Based Model (EBM), enabling it to actively identify and mitigate hazardous interactions caused by knowledge inapplicability. We validate the efficacy of this framework by deeply integrating it with CODA-Prompt, an advanced CL algorithm. Experiments demonstrate that KVM significantly reduces catastrophic failures caused by knowledge obsolescence without sacrificing learning efficiency, providing a rigorous solution for building safer and more reliable lifelong learning robotic systems.

## 1 Introduction

Lifelong Robot Learning aims to empower machines with the ability to continuously acquire and apply new skills in dynamic, unstructured physical environments, and is considered a key path toward general autonomous intelligence. To achieve this goal, the academic community has drawn upon numerous advancements from the field of Continual Learning (CL) (De Lange et al., 2021; Yang et al., 2025c;a; Kim et al., 2025a; Zheng et al., 2025a; Lai et al., 2025; Wang et al., 2025a; Wuerkaixi et al., 2025). Existing algorithms have achieved notable success in tackling the core challenge of "catastrophic forgetting" (McCloskey & Cohen, 1989; van de Ven et al., 2024; Cao et al., 2024; Ren et al., 2024; Luo et al., 2025b; Cong et al., 2025; Cho et al., 2025) through various strategies. These include experience replay (Lu et al., 2023; Neves et al., 2024; Li et al., 2024b; Hassani et al., 2025; Zhang et al., 2025a; Li et al., 2025a; Lu et al., 2025a; Lin et al., 2025), parameter isolation (Mallya & Lazebnik, 2018; Vicente-Sola et al., 2025; Zeng et al., 2025a; Bhat et al., 2025; Wang et al., 2025b; Saha et al., 2021; Zeng et al., 2025b; Zhu et al., 2025), and prompt-based learning with pre-trained models (Smith et al., 2023; Wang et al., 2022a; Lu et al., 2025b; Kang et al., 2025; Yang et al., 2025b; Li et al., 2025b; Chen et al., 2025; Le et al., 2024; Piao et al., 2024; Gao et al., 2024b).

These advanced CL algorithms are gradually being applied to robotics, showing potential in continual imitation learning (Zare et al., 2024; Le Mero et al., 2022; Zheng et al., 2022; Hua et al., 2021) and reinforcement learning (Wołczyk et al., 2021) tasks. However, a fundamental challenge emerges when migrating these algorithms, primarily validated in virtual environments, to physical systems. **Unlike in simulations, failed interactions in the physical world often entail high or even irreversible costs**, such as hardware damage, task failure, or safety threats to the surroundings (Gu et al., 2017; Dulac-Arnold et al., 2019). The root of these failures is the endogenous safety risks (Amodei et al., 2016; Gyevnar & Kasirzadeh, 2025) caused by policies that the agent itself has learned but which become inappropriate in new contexts.

Current CL work in robotics largely inherits the core objective from its virtual-world origins: the effective retention and accumulation of knowledge. This paradigmatic inertia means that a skill, once optimal in a specific context, may not just become "useless" after an environmental change.

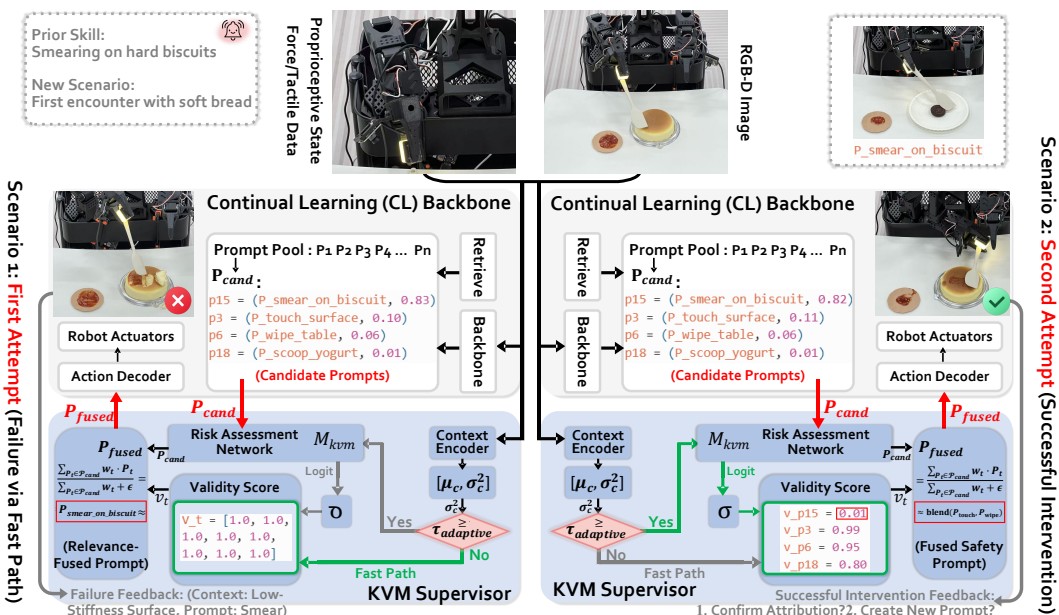

Figure 1: Overview of our proposed framework. The Continual Learning (CL) core (top) is responsible for retrieving a "proposal" from the prompt pool based on the current observation $x_t$. This proposal includes a set of candidate prompts $\mathcal{P}_{cand}$ and their relevance scores. The KVM supervisory module (bottom) receives this proposal in parallel and processes rich multimodal context information $C_{raw}$ for an independent risk assessment. Its internal "Smart Gate" mechanism ($\tau_{adaptive}$) dynamically decides whether to activate the full risk assessment network based on contextual uncertainty. Finally, KVM outputs a "consensus" prompt $P_{fused}$ that fuses "relevance" and "safety," which is returned to the action decoder to generate the final safe action $a_t$. The figure contrasts two scenarios, showing how KVM enables a safe interaction (Scenario 2) through successful real-time risk intervention after an initial failure due to knowledge obsolescence (Scenario 1).

Instead, it can become a direct cause of catastrophic failure. This forces the system into a **trial-and-error learning loop** that is costly and unacceptable in the physical world. This unique challenge, directly linked to physical risk and triggered by "Knowledge Obsolescence," remains a neglected research gap in lifelong robot learning. The existing CL paradigm focuses on "how to remember" but generally lacks a mechanism to actively manage "how to forget," which is a significant bottleneck for its application in real-world, high-stakes robotics.

To this end, inspired by memory inhibition theories from cognitive science, we introduce and define a new mechanism in lifelong robot learning: **Contextual Forgetting**. We posit that contextual forgetting—the ability to actively suppress an obsolete policy under specific conditions rather than permanently deleting it—is key to mitigating the risks of knowledge obsolescence. To implement this, we propose the Knowledge Validity Module (KVM), a modular Meta-cognitive Supervisory Layer. KVM operates independently of the core decision-making process of the CL algorithm, ensuring the objectivity of its risk assessment. In this work, we validate its effectiveness by integrating KVM with CODA-Prompt (Smith et al., 2023), an advanced prompt-based CL algorithm. Its core function is to perform real-time, independent risk assessments on "candidate knowledge" retrieved by the CL core. It then synthesizes a safety-vetted "consensus" knowledge representation and returns it to the CL core to guide the generation of a final, safe action.

This study selects the state-of-the-art rehearsal-free continual learning algorithm, CODA-Prompt (Smith et al., 2023), as our learning backbone. We integrate the KVM module with it and conduct experimental validation across a series of rigorous robotic manipulation tasks. Our main contributions include:

- We systematically articulate the endogenous safety risks in lifelong robot learning caused by high-confidence obsolete policies. We are the first to propose "Contextual Forgetting" as the core mechanism to address this problem.

- We design and implement a novel meta-cognitive functional module (KVM). Its core innovations include: (1) A principled risk assessment framework based on an **Energy-Based Model (EBM)**, which quantifies risk by learning the **energy landscape** of safe interactions. (2) A self-supervised closed loop based on **Contrastive Learning** to enhance the feature space's ability to discriminate between safe and hazardous interactions. (3) An evolution mechanism that includes "verification-based attribution" and "knowledge gap confirmation" to guide the safe evolution of the knowledge base. **We provide a detailed theoretical justification for this framework in Appendix §C.**

- Through extensive experiments in both simulated and real-world robotic tasks, we demonstrate that KVM significantly reduces catastrophic failures caused by "knowledge obsolescence" without compromising learning efficiency. This validates the critical value of our proposed framework in building safer, more reliable lifelong learning robotic systems.

## 2 RELATED WORK

Our research is positioned at the intersection of continual learning, robot learning, and safety. While dominant paradigms in Continual Learning (CL) aim to mitigate catastrophic forgetting and ensure knowledge **retention** (McCloskey & Cohen, 1989; De Lange et al., 2021), their strategies—ranging from experience replay (Chaudhry et al., 2019; Buzzega et al., 2020; Shin et al., 2017) and regularization (Kirkpatrick et al., 2017; Aljundi et al., 2018; Rusu et al., 2016; Mallya & Lazebnik, 2018) to prompt-based methods (Wang et al., 2022b;a; Smith et al., 2023)—are designed to effectively **archive** knowledge. When applied to robotics for continual imitation or reinforcement learning (Zeng et al., 2025c; Roy et al., 2025; Luo et al., 2025a; Wan et al., 2024; Lomonaco et al., 2020; Wołczyk et al., 2021; Kumar et al., 2025; Nayyar & Srivastava, 2025; Erden et al., 2025; Lozano-Cuadra et al., 2025; Malagon et al., 2025; Meng et al., 2025), this focus on "combating forgetting" overlooks a critical challenge in the physical world: the safety risks posed by **knowledge obsolescence**, where a once-correct policy becomes harmful.

Orthogonally, established work on robot safety (García & Fernández, 2015) primarily addresses risks from **insufficient knowledge** (i.e., the unknown). Safe Reinforcement Learning, through methods like constrained optimization (Achiam et al., 2017), and uncertainty estimation techniques (Kahn et al., 2017; Hafner et al., 2023), are designed to manage exploration in novel states where a model is uncertain. Our work tackles a different and more insidious problem: risks arising from **obsolete knowledge**, where a model acts with high confidence but is dangerously incorrect. We focus on identifying and intervening against these "confident errors," a dimension not explicitly covered by traditional safety approaches.

Thus, a critical research gap remains: how to systematically address the **endogenous safety risks arising from "knowledge obsolescence" in lifelong robot learning**. Our "Contextual Forgetting" framework is proposed to fill this gap, reframing "forgetting" from a passive failure to be overcome into an active, meta-cognitive function for managing high-confidence risks.

## 3 METHOD

To address endogenous safety risks from "knowledge obsolescence," we propose a meta-cognitive supervisory layer, the Knowledge Validity Module (KVM). KVM is designed to synergistically augment prompt-based Continual Learning (CL) frameworks, such as CODA-Prompt (Smith et al., 2023), which we use as our backbone. The core idea is to decouple risk assessment from the primary learning objective, as illustrated in Figure 1. A detailed theoretical justification for our framework is provided in Appendix §C.

### 3.1 THE KVM MODULE: RISK ASSESSMENT VIA ENERGY-BASED MODELS

The KVM module consists of a Context Encoder ($E_c$) and a KVM Supervisor ($M_{\text{kvm}}$) that operationalizes our core mechanism, **Contextual Forgetting**.

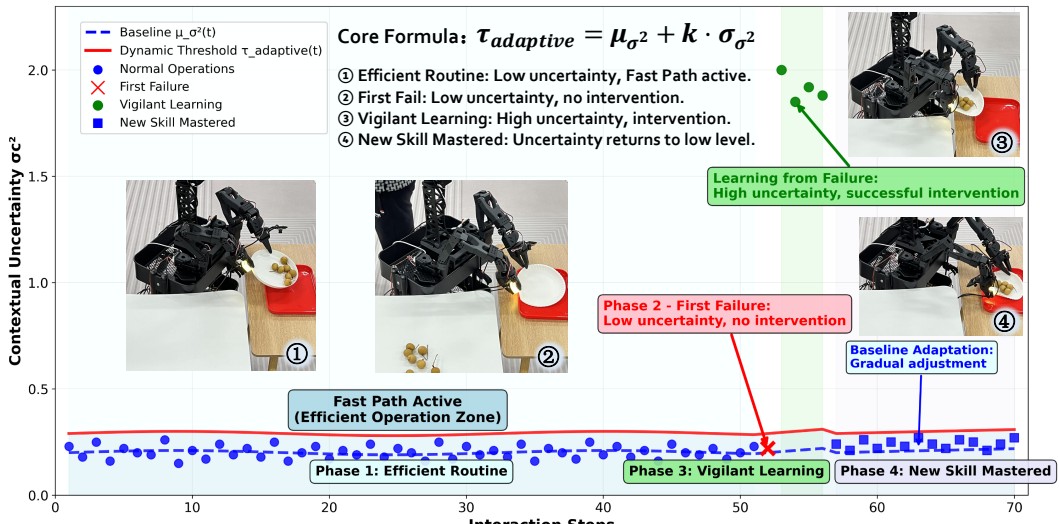

Figure 2: Operational mechanism of the Smart Gate ($\tau_{adaptive}$), illustrating how it ensures both efficiency and safety. The dynamic threshold's sensitivity factor k is set to 3 for the data shown. The graph plots contextual uncertainty ($\sigma_c^2$) over time, highlighting four key phases: (1) An efficient routine on the 'Fast Path' where uncertainty is low. (2) A failure occurs when outdated knowledge (picking up a plate from the nearer side) is misapplied to a new object (a disposable plate), but the gate correctly remains inactive due to low initial uncertainty. (3) The failure spikes uncertainty, activating the KVM for a successful, safe intervention. (4) Following successful interventions, uncertainty returns to baseline as the new skill (picking up the plate from both sides for stability) is mastered. This illustrates that the KVM remains computationally inexpensive by default.

**Context Encoder and Smart Gate.** The encoder $E_c$ maps raw multimodal context $c_{\text{raw}}$ into a Gaussian distribution $(\mu_c, \sigma_c^2)$. To enhance efficiency, an adaptive "Smart Gate" activates the full risk assessment only when contextual uncertainty $\sigma_c^2$ exceeds a dynamic threshold $\tau_{\text{adaptive}}$, which is statistically derived from recent safe interactions:

$$\tau_{\text{adaptive}} = \mu_{\sigma^2} + k \cdot \sigma_{\sigma^2} \tag{1}$$

In this equation, $\tau_{\text{adaptive}}$ is the activation threshold, $k$ is a sensitivity factor, and $\mu_{\sigma^2}$ and $\sigma_{\sigma^2}$ are the running mean and standard deviation of uncertainty from recent successful interactions.

**KVM Supervisor.** We formulate risk assessment using an **Energy-Based Model (EBM)** (Grathwohl et al., 2019), chosen for its ability to model complex, non-linear safety manifolds. The supervisor learns a neural energy function $E_\theta(\mathbf{h})$ that assigns a scalar energy value to a given interaction feature representation $\mathbf{h}$. This energy is normalized into a validity score $v_t \in (0, 1]$:

$$v_t = \frac{1}{1 + e^{-\gamma(-E_\theta(\mathbf{h}_t) - l_0')}} \tag{2}$$

Here, $v_t$ is the validity score for the feature vector $\mathbf{h}_t$ at timestep $t$. $E_\theta(\mathbf{h}_t)$ is the energy computed by the EBM with parameters $\theta$. The hyperparameters $\gamma$ and $l_0'$ control the steepness and center of the Sigmoid function, respectively, to calibrate risk sensitivity.

### 3.2 IMPLEMENTATION OF CONTEXTUAL FORGETTING

KVM implements Contextual Forgetting through immediate risk mitigation and long-term knowledge base optimization.

**Real-time Risk Mitigation.** The system balances the CL core's "relevance" with KVM's "safety" assessment. We define a final decision weight $w_t = v_t \cdot \alpha_t$, where $\alpha_t$ is the relevance score from the CL core and $v_t$ is the safety score from KVM. This weight ensures that only prompts that are

both relevant and safe contribute significantly to the fused "consensus" prompt $P_{\text{fused}}$. This prompt is then used by the decoder to generate the final action $a_{\text{final}}$.

$$P_{\text{fused}} = \frac{\sum_{P_t \in \mathcal{P}_{\text{cand}}} w_t \cdot P_t}{\sum_{P_t \in \mathcal{P}_{\text{cand}}} w_t + \epsilon} \tag{3}$$

$$a_{\text{final}} = f_{\text{decoder}}(f_{\text{backbone}}(x_t), P_{\text{fused}}) \tag{4}$$

In these equations, $\mathcal{P}_{\text{cand}}$ is the set of candidate prompts proposed by the CL core based on the current observation $x_t$, and $\epsilon$ is a small constant for numerical stability. $f_{\text{backbone}}$ and $f_{\text{decoder}}$ are the backbone network and action decoder, respectively.

**Knowledge Base Metabolism.** To ensure long-term adaptability, KVM also incorporates a mechanism to track the long-term performance of each prompt and prune skills that become chronically obsolete. The specific metrics are detailed in Appendix §A.1.

### 3.3 KVM's Self-Supervised Learning and Knowledge Evolution

KVM's capabilities evolve through a self-supervised process triggered by failures. This process is driven by two synergistic losses that jointly optimize the feature space and the energy model.

First, a contrastive loss, $\mathcal{L}_{\text{contrast}}$, shapes a feature space where safe and hazardous interactions are geometrically separable:

$$\mathcal{L}_{\text{contrast}} = \mathbb{E}[\max(0, D(\mathbf{h}_{\text{anchor}}, \mathbf{h}_{\text{safe}}) - D(\mathbf{h}_{\text{anchor}}, \mathbf{h}_{\text{fail}}) + m)] \tag{5}$$

Here, $\mathbb{E}$ is the expectation over training samples, $D(\cdot, \cdot)$ is the Euclidean distance, and $m$ is a margin hyperparameter. The terms $\mathbf{h}_{\text{anchor}}$, $\mathbf{h}_{\text{safe}}$, and $\mathbf{h}_{\text{fail}}$ represent the feature vectors for an anchor context, a corresponding safe interaction, and a hazardous interaction, respectively.

Second, an energy contrastive loss, $\mathcal{L}_{\text{energy}}$, trains the EBM to map safe samples to low energy and hazardous samples to high energy:

$$\mathcal{L}_{\text{energy}} = \mathbb{E}_{\mathbf{h}_{\text{safe}} \sim \mathcal{D}_{\text{safe}}}[E_\theta(\mathbf{h}_{\text{safe}})] + \mathbb{E}_{\mathbf{h}_{\text{fail}} \sim \mathcal{D}_{\text{fail}}}[\max(0, m_e - E_\theta(\mathbf{h}_{\text{fail}}))] \tag{6}$$

In this loss, expectations are taken over safe samples ($\mathbf{h}_{\text{safe}}$) from the safe experience set $\mathcal{D}_{\text{safe}}$ and hazardous samples ($\mathbf{h}_{\text{fail}}$) from the hazardous set $\mathcal{D}_{\text{fail}}$. The term $m_e$ is a margin that penalizes low-energy hazardous samples. This entire learning process is fueled by a verification-based attribution mechanism that identifies the root cause of failures to provide high-quality negative samples and guide knowledge evolution, as detailed in Appendix §D.

## 4 Experiments

We conducted a series of experiments to validate the effectiveness of our proposed Knowledge Validity Module (KVM). Our evaluation demonstrates KVM's performance on complex tasks (Q1), its robustness to domain shifts (Q2), the function of its core mechanisms (Q3), and the contribution of its key components (Q4).

### 4.1 Experimental Setup

**Benchmarks.** To comprehensively evaluate our method, we tested it on two widely adopted and challenging robotic manipulation benchmarks: **LIBERO** (Liu et al., 2023), a primary benchmark for assessing generalization across multiple dimensions (spatial layout, objects, goals, task length), and **SimplerEnv** (Li et al., 2025c), which is used to evaluate model robustness against significant domain shifts (e.g., changes in lighting, textures, and camera angles).

**Implementation Details.** Our core method, **CODA-Prompt + KVM (Ours)**, integrates our proposed KVM module with the advanced continual learning algorithm CODA-Prompt (Smith et al., 2023). For fair comparison, we directly cite publicly available performance data from recent related works on these benchmarks. Our model is trained only on the datasets specified by the respective benchmarks, without using any additional large-scale external datasets. For a detailed analysis of the computational overhead introduced by our method, please see Appendix H.2.

**"Knowledge Obsolescence" Challenge Scenarios.** To specifically evaluate the core capability of our "Contextual Forgetting" mechanism in addressing endogenous safety risks from "knowledge obsolescence," we designed and implemented a comprehensive benchmark of 18 challenging scenarios, with representative scenarios shown in Figure 4. These scenarios simulate common failure modes caused by changes in physical properties, oversight of interaction preconditions, or misleading representational similarity. As shown in Table 1, we compare our method against various mainstream VLA models on this benchmark. Detailed descriptions of all scenarios are provided in Appendix B.

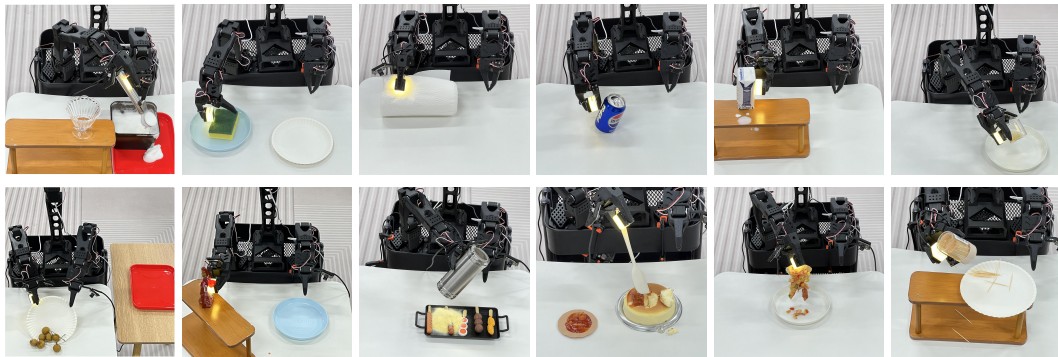

Figure 3: An overview of the challenging scenarios in our custom "Knowledge Obsolescence" benchmark. These tasks are designed to evaluate the robot's ability to identify and adapt to situations where prior knowledge becomes outdated and potentially hazardous due to changes in object properties, interaction preconditions, or misleading perceptual similarities.

## 4.2 RESULTS ON MAIN ROBOTIC BENCHMARKS (Q1 & Q2)

To answer Q1 and Q2, we compared our method against current state-of-the-art techniques on the LIBERO and SimplerEnv benchmarks.

**LIBERO Experimental Results.** As shown in Table 2, our method performs exceptionally well across all four evaluation suites, achieving an average success rate of 86.2%. Particularly on the *Long* horizon tasks, our performance shows a significant improvement over strong baselines like UniVLA. We attribute this to KVM's "Contextual Forgetting" mechanism, which actively suppresses early-stage policies that may become unsuitable later in long-sequence tasks. This effectively avoids potential catastrophic failures and improves the end-to-end task success rate.

**SimplerEnv Experimental Results.** In Table 3, we demonstrate the model's robustness against significant visual domain shifts. Our method achieves an average success rate of 70.3%, surpassing baselines including UniVLA on several tasks. This validates that KVM, as a meta-cognitive supervisory layer, can effectively identify "cognitive uncertainty" and "knowledge inapplicability" caused by domain shifts. It generates safer, more robust actions by dynamically adjusting policy weights.

## 4.3 QUALITATIVE ANALYSIS OF REAL-WORLD FAILURE CASES (Q3)

To illustrate how our mechanism handles failures from "knowledge obsolescence" (Q3), we analyze two representative cases. The first ("stuck drawer") highlights real-time physical risk avoidance, while the second ("pouring honey") showcases KVM's unique diagnostic capabilities compared to traditional OOD detection.

**Case 1: Precondition Oversight ("Stuck Drawer").** This case shows KVM's response to physical feedback. When an agent encounters a stuck drawer visually identical to previously smooth ones, a standard VLA would apply continuous, damaging force. In contrast, KVM's context encoder immediately detects the abnormal "high resistance, zero displacement" state from force/torque sensors. This significant deviation from the safe interaction manifold leads the energy model $E_\theta(\mathbf{h})$ to assign a high energy value, which is converted to a low validity score, preemptively suppressing the hazardous policy before damage occurs.

**Case 2: Policy Failure and Self-Supervised Evolution ("Pouring Honey").** This case clarifies KVM's advantage over OOD detection. An agent proficient in "pouring water" will fail when en-

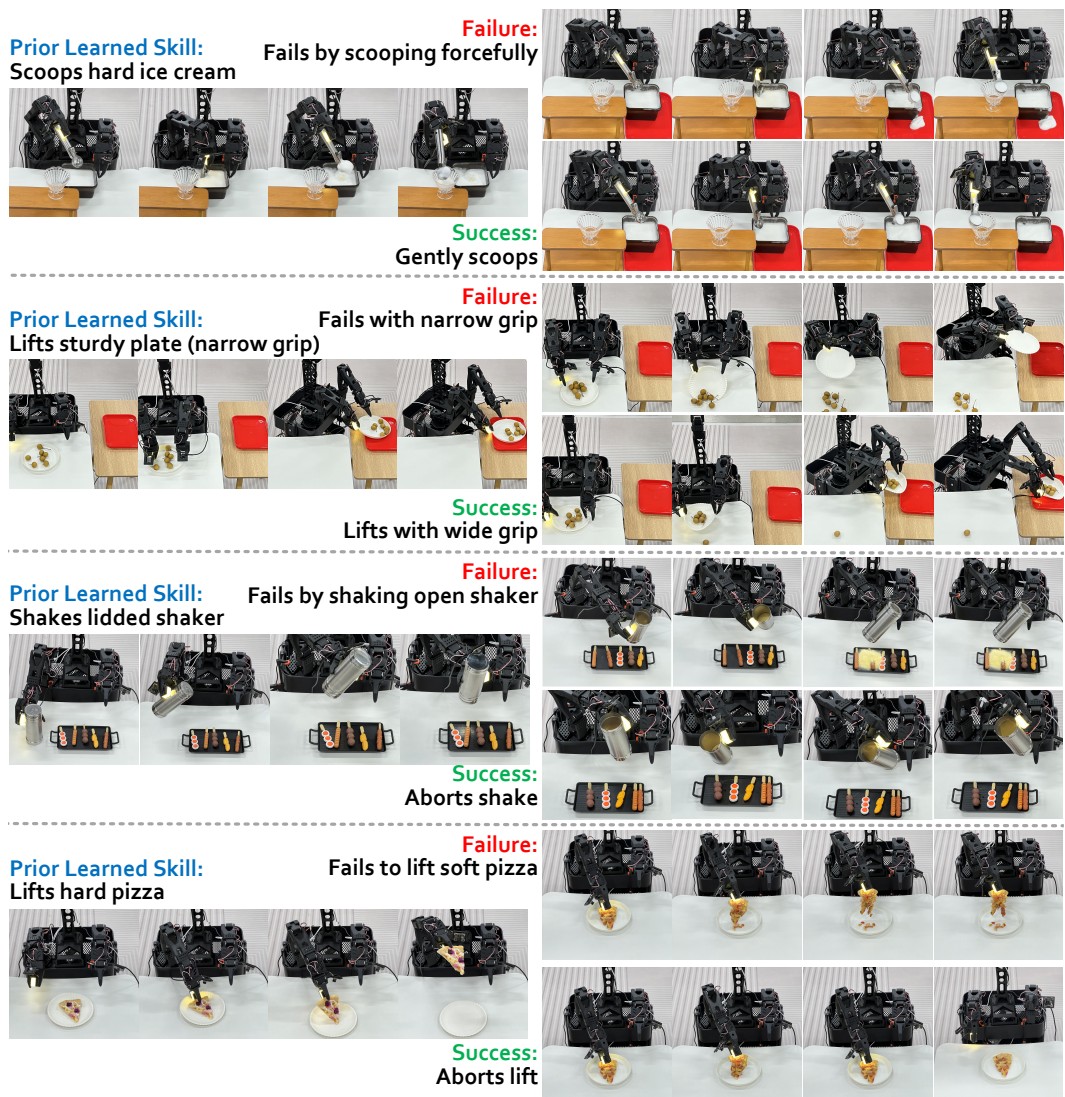

Figure 4: Qualitative results from our "Knowledge Obsolescence" benchmark. Each of the four rows showcases a complete sequence: the robot first applies a **Prior Learned Skill**; then, it **Fails** on its first attempt in a new, deceptively similar scenario; finally, it **Succeeds** on the second attempt through our method's intervention, demonstrating effective adaptation.

countering high-viscosity honey by misapplying its obsolete knowledge. While an OOD system might detect a statistical anomaly (e.g., "low-probability task progress"), this generic signal is insufficient for targeted policy correction.

KVM's approach is fundamentally different. It performs a **knowledge-driven consistency check** on the proposed "pouring water" prompt. Upon detecting a persistent discrepancy between the prompt's implicit physical model and the observed slow flow rate, KVM attributes the failure directly to the knowledge unit, leading to the introspective diagnosis: **"The 'pouring water' knowledge model is wrong in this context."**

This precise credit assignment is crucial for adaptation. It allows KVM to generate a targeted negative sample ($\mathbf{h}_{\text{fail}}$) to update its energy model and signal the CL core to create a new, more appropriate skill, as detailed in Appendix §D. This process transforms a failure into a generalizable cognitive enhancement.

Table 1: **Performance on the "Knowledge Obsolescence" Challenge Benchmark.** We report the success rate (%) across three categories of challenge scenarios. Each task is executed twice in succession (the first attempt is direct, the second is based on gradient feedback or KVM intervention). This process is repeated 10 times (resetting to pre-trained initial weights each time), and the average success rate is reported. Traditional VLA models perform poorly in these scenarios designed to test "knowledge obsolescence," whereas our method demonstrates superior risk identification and avoidance capabilities.

| Model | Params(B) | A. Phys. Mismatch (%) | B. Precond. Oversight (%) | C. Repr. Sim. (%) | Avg. (%) |
|---|---|---|---|---|---|
| FlowVLA (Zhong et al., 2025) | 8.5 | 2.8/5 (56.0) | 2.0/5 (40.0) | 3.7/8 (46.2) | 8.5/18 (47.2) |
| UnifiedVLA (Wang et al., 2025c) | 8.5 | 2.2/5 (44.0) | 2.9/5 (58.0) | 4.6/8 (57.5) | 9.7/18 (53.9) |
| OpenVLA (Kim et al., 2024) | 7 | 0.9/5 (18.0) | 1.9/5 (38.0) | 1.7/8 (21.2) | 4.5/18 (25.0) |
| OpenVLA-OFT (Kim et al., 2025b) | 7 | 1.8/5 (36.0) | 3.0/5 (60.0) | 3.7/8 (46.2) | 8.5/18 (47.2) |
| UniVLA (Bu et al., 2025) | 7 | 1.8/5 (36.0) | 1.9/5 (38.0) | 2.5/8 (31.2) | 6.2/18 (34.4) |
| CoT-VLA (Zhao et al., 2025) | 7 | 2.7/5 (54.0) | 1.8/5 (36.0) | 2.4/8 (30.0) | 6.9/18 (38.3) |
| WorldVLA (Cen et al., 2025) | 7 | 3.7/5 (74.0) | 0.9/5 (18.0) | 1.9/8 (23.8) | 6.5/18 (36.1) |
| TraceVLA (Zheng et al., 2025b) | 7 | 0.7/5 (14.0) | 1.0/5 (20.0) | 2.9/8 (36.2) | 4.6/18 (25.6) |
| MolmoAct (Lee et al., 2025) | 7 | 2.7/5 (54.0) | 2.9/5 (58.0) | 3.5/8 (43.8) | 9.1/18 (50.6) |
| ThinkAct (Huang et al., 2025) | 7 | 2.0/5 (40.0) | 1.8/5 (36.0) | 4.3/8 (53.8) | 8.1/18 (45.0) |
| PD-VLA (Song et al., 2025) | 7 | 2.5/5 (50.0) | 1.7/5 (34.0) | 3.9/8 (48.8) | 8.1/18 (45.0) |
| 4D-VLA (Zhang et al., 2025b) | 4 | 1.7/5 (34.0) | 2.6/5 (52.0) | 2.8/8 (35.0) | 7.1/18 (39.4) |
| SpatialVLA (Qu et al., 2025) | 4 | 0.9/5 (18.0) | 1.8/5 (36.0) | 3.3/8 (41.2) | 6.0/18 (33.3) |
| $\pi_0$ (Black et al., 2024) | 3 | 1.5/5 (30.0) | 1.9/5 (38.0) | 3.0/8 (37.5) | 6.4/18 (35.6) |
| $\pi_0$-FAST (Pertsch et al., 2025) | 3 | 0.7/5 (14.0) | 0.8/5 (16.0) | 1.5/8 (18.8) | 3.0/18 (16.7) |
| NORA (Hung et al., 2025) | 3 | 1.4/5 (28.0) | 2.7/5 (54.0) | 1.8/8 (22.5) | 5.9/18 (32.8) |
| SmolVLA (Shukor et al., 2025) | 2.2 | 1.8/5 (36.0) | 1.6/5 (32.0) | 1.7/8 (21.2) | 5.1/18 (28.3) |
| GR00T N1 (NVIDIA et al., 2025) | 2 | 0.6/5 (12.0) | 1.7/5 (34.0) | 2.3/8 (28.7) | 4.6/18 (25.6) |
| GraspVLA (Deng et al., 2025) | 1.8 | 2.4/5 (48.0) | 0.7/5 (14.0) | 0.9/8 (11.2) | 4.0/18 (22.2) |
| Seer[†] (Tian et al., 2025) | 0.57 | 0.5/5 (10.0) | 0.9/5 (18.0) | 1.3/8 (16.2) | 2.7/18 (15.0) |
| VLA-OS (Gao et al., 2025) | 0.5 | 1.3/5 (26.0) | 1.5/5 (30.0) | 0.7/8 (8.8) | 3.5/18 (19.4) |
| Diffusion Policy[†] (Chi et al., 2023) | - | 0.4/5 (8.0) | 0.6/5 (12.0) | 0.6/8 (7.5) | 1.6/18 (8.9) |
| CODA-Prompt (Backbone) (Smith et al., 2023) | 7 | 0.7/5 (14.0) | 1.4/5 (28.0) | 1.3/8 (16.2) | 3.4/18 (18.9) |
| **CODA-Prompt + KVM (Ours)** | **7** | **4.8/5 (96.0)** | **4.9/5 (98.0)** | **6.7/8 (83.8)** | **16.4/18 (91.1)** |

Table 2: **Results on the LIBERO benchmark.** We report the final task success rate (**%**). We compare our method against existing state-of-the-art methods. Notably, our approach achieves competitive performance **without relying on massive external datasets**.

| Model | Large Scale Pretrain | Spatial | Object | Goal | Long | **Avg.** |
|---|---|---|---|---|---|---|
| *w/o World Model* | | | | | | |
| Diffusion Policy (Chi et al., 2023) | × | 78.3 | 92.5 | 68.3 | 50.5 | 72.4 |
| Octo (Ghosh et al., 2024) | ✓ | 78.9 | 85.7 | 84.6 | 51.1 | 75.1 |
| OpenVLA (Kim et al., 2024) | ✓ | 84.7 | 88.4 | 79.2 | 53.7 | 76.5 |
| DiT Policy (Hou et al., 2025) | ✓ | 84.2 | 96.3 | 85.4 | 63.8 | 82.4 |
| TraceVLA (Zheng et al., 2025b) | ✓ | 84.6 | 85.2 | 75.1 | 54.1 | 74.8 |
| SpatialVLA (Qu et al., 2025) | ✓ | 88.2 | 89.9 | 78.6 | 55.5 | 78.1 |
| pi0-FAST (Pertsch et al., 2025) | ✓ | 96.4 | 96.8 | 88.6 | 60.2 | 85.5 |
| ThinkAct (Huang et al., 2025) | ✓ | 88.3 | 91.4 | 87.1 | 70.9 | 84.4 |
| *w/ World Model* | | | | | | |
| WorldVLA (Cen et al., 2025) | × | 85.6 | 89.0 | 82.6 | 59.0 | 79.1 |
| UniVLA[†] (Wang et al., 2025c) | × | 92.6 | 93.8 | 86.6 | 63.0 | 84.0 |
| CoT-VLA (Zhao et al., 2025) | ✓ | 87.5 | 91.6 | 87.6 | 69.0 | 81.1 |
| **CODA-Prompt + KVM (Ours)** | × | **92.8** | **94.1** | **87.5** | **70.5** | **86.2** |

† UniVLA results are from its official implementation, trained only on LIBERO without a wrist camera for a fair comparison.

## 4.4 ABLATION STUDY (Q4)

Finally, we conducted a series of ablation studies to evaluate the contribution of each key component within the KVM framework. As shown in Table 4, all evaluations were performed on the more challenging LIBERO-Long benchmark. We report both task success rate and catastrophic failure rate (CFR).

Table 3: **Results on the SimplerEnv-WidowX benchmark.** We report the final task success rate (%). Our method demonstrates strong robustness to the visual domain shifts present in this benchmark.

| Model | Put Spoon | Put Carrot | Stack Block | Put Eggplant | **Avg.** |
|---|---|---|---|---|---|
| RT-1-X (Ghosh et al., 2024) | 0.0 | 4.2 | 0.0 | 0.0 | 1.1 |
| Octo-Base (Ghosh et al., 2024) | 12.5 | 8.3 | 0.0 | 43.1 | 16.0 |
| Octo-Small (Ghosh et al., 2024) | 47.2 | 9.7 | 4.2 | 56.9 | 29.5 |
| OpenVLA (Ghosh et al., 2024) | 0.0 | 0.0 | 0.0 | 4.1 | 1.0 |
| RoboVLMs (Li et al., 2024a) | 45.8 | 20.8 | 4.2 | 79.2 | 37.5 |
| SpatialVLA (Qu et al., 2025) | 16.7 | 25.0 | 29.2 | 100 | 42.7 |
| RoboPoint (Yuan et al., 2024) | 16.7 | 20.8 | 8.3 | 25.0 | 17.7 |
| FSD (Yuan et al., 2025a) | 41.6 | 50.0 | 33.3 | 37.5 | 40.6 |
| Embodied-R1 (Yuan et al., 2025b) | 62.5 | **68.0** | 36.1 | 58.3 | 56.2 |
| UniVLA[†] (Wang et al., 2025c) | 62.5 | 62.5 | 41.6 | 95.8 | 65.6 |
| **CODA-Prompt + KVM (Ours)** | **65.0** | 63.0 | **55.0** | **98.0** | **70.3** |

† UniVLA results were obtained by evaluating its officially released checkpoint.

The results clearly show that removing the entire KVM module (reverting to the CODA-Prompt baseline) causes the catastrophic failure rate to surge by over 15 percentage points. This confirms our framework's core value in enhancing system safety.

Next, we ablated KVM's two core components: (1) **w/o Smart Gate**: We removed the adaptive activation mechanism. The results show that while this had little impact on the success rate, it validates that the "Smart Gate" effectively avoids unnecessary computational overhead in most safe, routine situations. (2) **w/o Energy Model**: We replaced the energy-based risk assessment with a simple MLP binary classifier trained with standard binary cross-entropy (BCE) loss. The results show a significant performance drop and a sharp increase in the failure rate. This highlights the superiority of our energy-based framework, which learns the energy landscape of safe interactions and can better generalize to unseen, unexpected hazardous situations—a feat difficult for standard discriminative classifiers.

Table 4: **Ablation study of KVM components on the LIBERO-Long benchmark.** We evaluate the impact of removing key KVM designs on task success rate (%) and catastrophic failure rate (CFR, %).

| Configuration | Success Rate (%) | Catastrophic Failure Rate (%) |
|---|---|---|
| **CODA-Prompt + KVM (Full Model)** | **70.5** | **4.7** |
| *Ablations:* | | |
| 1. w/o KVM (Baseline) | 68.6 | 19.8 |
| 2. w/o Smart Gate | 70.1 | 5.1 |
| 3. w/o Energy Model | 62.5 | 14.3 |

# 5 CONCLUSION

To address the critical safety risks arising from "knowledge obsolescence" in lifelong robot learning, this research introduces "Contextual Forgetting," a novel mechanism implemented via a modular meta-cognitive supervisor, KVM. Our experiments demonstrate that KVM significantly reduces catastrophic failures caused by obsolete knowledge without sacrificing task performance. This work thus provides a practical solution for building safer lifelong learning robots and, more importantly, expands the focus of continual learning from merely "how to remember" to the equally critical dimension of "how to forget safely." Future work could focus on distinguishing high-risk situations from benign novelty to enable more efficient safe exploration.

## ETHICS STATEMENT

This work does not raise any direct ethical issues. Our research aims to improve the safety of lifelong learning robotic systems by preventing catastrophic failures, which we believe is a positive ethical contribution.

## REPRODUCIBILITY STATEMENT

We are committed to ensuring the full reproducibility of our research. Below, we outline the detailed resources and specific information provided to support this goal.

- **Code Availability:** The complete source code for our KVM module, including the context encoder ($E_c$), the energy-based supervisor ($M_{kvm}$), and the integration framework with the CODA-Prompt backbone, will be made publicly available in an anonymized repository upon publication. Our implementation supports CUDA 12.4. All experimental scripts, training loops, evaluation protocols, and data preprocessing pipelines are included to facilitate complete replication.

- **Datasets and Benchmarks:** Our experiments utilize both established and custom benchmarks:
  - **LIBERO** (Liu et al., 2023): We use all four evaluation suites (Spatial, Object, Goal, Long) with the official data splits and evaluation protocols.
  - **SimplerEnv** (Li et al., 2025c): We evaluate on the WidowX manipulation tasks with the standard domain shift configurations.
  - **Knowledge Obsolescence Benchmark**: We introduce 18 carefully designed challenge scenarios categorized into three types: Physical Property Mismatch (5 scenarios), Interaction Precondition Oversight (5 scenarios), and Representational Similarity Misleading (8 scenarios). Complete scenario descriptions, environmental setups, success criteria, and failure conditions are provided in Appendix §B.

- **Model Architecture and Training Details:** We provide comprehensive implementation details:
  - (1) Network architectures for all components (context encoder, energy function, integration layers) in Appendix §E.
  - (2) Complete hyperparameter specifications including learning rates, batch sizes, and optimization schedules.
  - (3) Training procedures for the three-stage learning process (contrastive feature learning, energy function training, and knowledge evolution).
  - (4) Evaluation metrics and statistical significance testing procedures.

- **Computational Environment:** All experiments were conducted on the hardware configuration detailed in Table 5. Upon publication, we will also provide:
  - (1) Complete software environment specifications.
  - (2) Docker container configurations for exact environment replication.
  - (3) Expected training times and computational requirements for each experiment.
  - (4) Random seeds and initialization procedures to ensure deterministic results.

Table 5: Hardware and Software Configuration for Experiments.

| Component | Specification |
|---|---|
| GPU | $4 \times$ NVIDIA A100-SXM4-80GB |
| CPU | Intel(R) Xeon(R) Platinum 8358P @ 2.60GHz (64 Cores) |
| Memory | 314 GiB |
| Storage | 1 TB |
| OS | Ubuntu 22.04.3 LTS |
| CUDA Version | 12.4 |

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

## A CORE FORMULA AND METRIC DEFINITIONS

### A.1 KNOWLEDGE BASE PRUNING METRIC DEFINITIONS

As described in Section §3.2, the knowledge base metabolism mechanism relies on two core long-term performance metrics: the "Failure Contribution Rate in New Contexts" (FCRNC) and the "Suppression Rate in New Contexts" (SRNC). Their specific mathematical definitions are as follows:

$$\text{FCRNC}(P_i) = \frac{\sum_{j \in \mathcal{I}_i} \mathbb{I}(o_j = \text{failure})}{|\mathcal{I}_i|} \tag{7}$$

$$\text{SRNC}(P_i) = \frac{\sum_{j \in \mathcal{I}_i} \mathbb{I}(v_j < \tau_v)}{|\mathcal{I}_i|} \tag{8}$$

Here, $\mathcal{I}_i$ is the set of indices for all interaction events where prompt $P_i$ was proposed by the CL core within a sliding window covering the N most recent tasks. $o_j \in \{\text{success}, \text{failure}\}$ is the final task outcome of the $j$-th interaction. $\mathbb{I}(\cdot)$ is the indicator function, which is 1 if the condition is true and 0 otherwise. $v_j$ is the validity score calculated by KVM for prompt $P_i$ in the $j$-th interaction, and $\tau_v$ is a low-score threshold used to identify a "suppression" action (e.g., $\tau_v = 0.1$).

## B   DETAILED "KNOWLEDGE OBSOLESCENCE" CHALLENGE SCENARIOS

To systematically validate the effectiveness of our framework, we constructed a series of challenge scenarios covering different physical properties and interaction types. These scenarios collectively simulate a core failure mode caused by "knowledge obsolescence": a previously effective, high-confidence policy becomes inapplicable or even harmful due to a failure to adapt to a subtle but critical change in the context, ultimately leading to catastrophic failure.

### B.1   A. PHYSICAL PROPERTY MISMATCH

The root cause of this failure category is the robot's erroneous generalization assumptions about an object's intrinsic, static physical properties, such as fragility, rigidity, hardness, and weight.

1. **Ice Cream (Frozen vs. Melted):**
   - *Outdated Knowledge:* Learned to scoop hard, frozen ice cream using a high-pressure, twisting motion.
   - *Failure Case:* When faced with a tub of partially melted, very soft ice cream, applying the same forceful strategy causes the scoop to lose control, flinging ice cream everywhere.

2. **Sponge (Dry vs. Wet):**
   - *Outdated Knowledge:* Learned to pick up a dry, lightweight sponge with minimal force.
   - *Failure Case:* When needing to lift a heavy, water-saturated sponge, applying the same low-force grasping policy results in failure to lift or dropping it mid-task.

3. **Potato (Raw vs. Cooked):**
   - *Outdated Knowledge:* Learned to handle a firm, raw potato with a strong grip.
   - *Failure Case:* When faced with a cooked, very soft potato, applying the same strong grip crushes it instantly.

4. **Clothing (Dry vs. Wet):**
   - *Outdated Knowledge:* Learned to pick up a dry, lightweight T-shirt with very little force.
   - *Failure Case:* When it needs to pick up a heavy, water-soaked garment from a washing machine, applying the same low-force grasping strategy is insufficient to lift it.

5. **Pizza (Cold vs. Hot):**
   - *Outdated Knowledge:* Learned to pick up a rigid, cooled slice of pizza by pinching its crust with one hand.
   - *Failure Case:* When faced with a fresh-out-of-the-oven, soft, non-rigid slice, pinching only the crust causes the front half to fold over, spilling all the toppings.

### B.2   B. OVERSIGHT OF INTERACTION PRECONDITION

The root cause of this failure category is the robot, due to over-generalization, completely ignoring whether a necessary "precondition" for a safe or effective interaction is met.

6. **Container (Sealed vs. Open):**
   - *Outdated Knowledge:* Learned it can securely grasp a "sealed" milk carton from the side.
   - *Failure Case:* By failing to notice the change in the "sealed" precondition (it has been opened), applying the old strategy squeezes the liquid out.

7. **Pepper Shaker (Capped vs. Uncapped):**
   - *Outdated Knowledge:* Learned to interact with a cap that has a "perforated filtering" function.
   - *Failure Case:* By failing to notice the change in the "filtering" precondition (the cap has been removed), applying the old strategy pours out all the pepper.

8. **Toothpick Dispenser (Small Hole vs. Large Opening):**

   - *Outdated Knowledge:* Learned to interact with a small hole that provides a "restricted outflow" function.
   - *Failure Case:* By failing to notice the change in the "restricted" precondition (the sliding cover is open), applying the old strategy causes all toothpicks to spill.

9. **Chip Bag (Unopened vs. Opened):**

   - *Outdated Knowledge:* Learned to interact with a bag that has an "airtight cushioning" function.
   - *Failure Case:* By failing to notice the change in the "cushioning" precondition (the bag is open), applying the old strategy crushes the chips.

10. **Drawer (Smooth vs. Stuck):**

    - *Outdated Knowledge:* Learned to interact with a drawer that has a "smooth sliding" function.
    - *Failure Case:* By failing to notice the change in the "smooth sliding" precondition (it is stuck), applying the old strategy leads to dangerous, continuous force application.

### B.3   C. MISLEADING REPRESENTATIONAL SIMILARITY

The root cause of this failure category is the robot encountering a new, different object that, due to high similarity in its feature representation to an old object, leads to a catastrophic knowledge transfer.

11. **Stone vs. Egg:**

    - *Outdated Knowledge:* Learned to grasp a hard stone with high force.
    - *Failure Case:* When faced with a fragile egg, which is visually and geometrically similar to a stone but fundamentally different, the robot wrongly transfers the "grasp stone" policy, crushing the egg.

12. **Book (Hardcover vs. Paperback):**

    - *Outdated Knowledge:* Learned to handle a sturdy hardcover book with a quick, forceful "push" motion.
    - *Failure Case:* When faced with a paperback of the same size but different material, the robot, misled by the high-level "book" representation, wrongly transfers the "push hardcover" policy, damaging the paperback's cover.

13. **Cracker vs. Soft Bread (Spreading):**

    - *Outdated Knowledge:* Learned to spread on a hard cracker using significant downward pressure.
    - *Failure Case:* When faced with "soft bread," which also requires spreading but is made of a completely different material, the robot wrongly transfers the high-force spreading policy due to task similarity, crushing the bread.

14. **Writing Tool (Ballpoint Pen vs. Brush):**

    - *Outdated Knowledge:* Learned that using a ballpoint pen requires applying continuous downward pressure.
    - *Failure Case:* When faced with a "brush," which is also a "writing tool" but has completely different physical properties, the robot wrongly transfers the "use pen" policy, leading to failure.

15. **Paper Towel Roll (With vs. Without Core):**

    - *Outdated Knowledge:* Learned to transport a "roll of paper" by grasping it from the hard cardboard core in the center.
    - *Failure Case:* When faced with a visually similar but structurally different "coreless paper towel roll," the robot wrongly transfers the grasping policy, crushing the roll.

16. **Plate (Sturdy vs. Disposable):**

- *Outdated Knowledge:* Learned to handle a "plate" by supporting its edges from both sides.
- *Failure Case:* When faced with a "disposable paper plate," which is also a "plate" but made of a completely different material, it wrongly transfers the policy designed for a sturdy plate, leading to failure.

17. **Soda Bottle (Full vs. Empty):**
   - *Outdated Knowledge:* Learned to push a stable, full "soda bottle."
   - *Failure Case:* When faced with an identical-looking "empty soda bottle," whose dynamics are drastically different due to a change in center of gravity and weight, the robot, misled by the highly similar "representation," wrongly transfers the pushing policy, causing it to tip over.

18. **Pouring Liquid (Water vs. Honey):**
   - *Outdated Knowledge:* Learned a policy for pouring a liquid like "water."
   - *Failure Case:* When faced with "honey," which is also a "liquid" but has completely different physical properties (viscosity), the robot wrongly transfers the water-pouring policy, leading to failure.

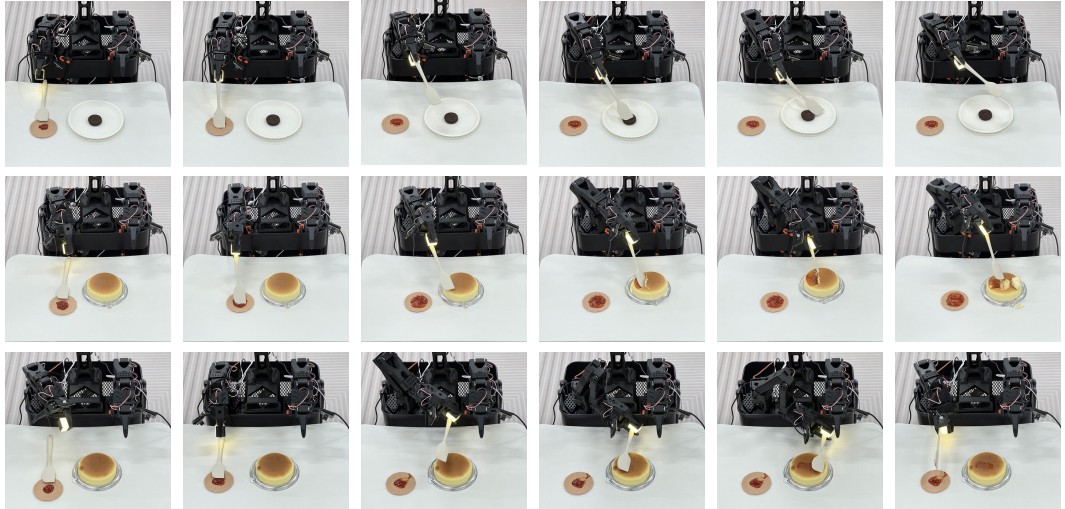

Figure 5: **Case Study on "Knowledge Obsolescence": Smearing on a Cracker vs. Soft Bread.** This figure qualitatively demonstrates how our KVM framework rapidly learns and adapts from a failure caused by "knowledge obsolescence." **(First Row) Prior Knowledge:** The agent has learned to apply significant downward pressure to spread on a hard cracker. **(Second Row) Initial Failure:** When faced with soft bread, the agent wrongly transfers the high-confidence outdated policy, applying excessive force that crushes the bread, resulting in task failure. **(Third Row) Success after KVM Intervention:** Following the failure, our KVM module successfully identifies and suppresses the high-risk outdated policy. On the next attempt, the agent adjusts its force and successfully completes the spreading task on the soft bread.

## C    THEORETICAL JUSTIFICATION FOR THE ENERGY-BASED RISK ASSESSMENT FRAMEWORK

This section provides a theoretical justification for the Energy-Based Model (EBM) risk assessment framework proposed in §3, and discusses its core assumptions and potential limitations.

### C.1    PROBLEM FORMULATION: ENERGY-BASED RISK

We formulate the problem of "knowledge obsolescence" risk in lifelong robot learning as the task of **online learning of a safe interaction energy function**. Let $\mathcal{H}$ be the joint feature space mapped by

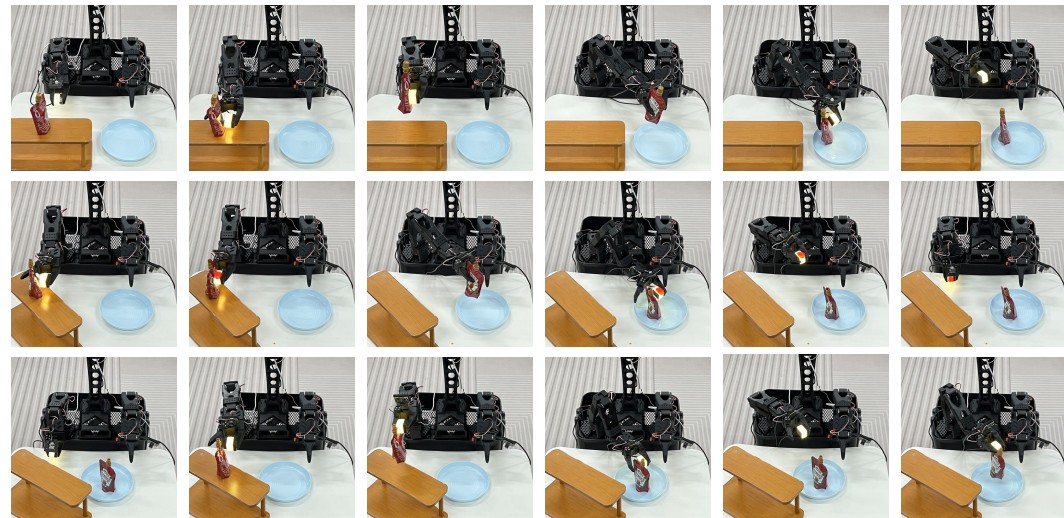

Figure 6: **Case Study on "Oversight of Interaction Precondition": Sealed vs. Open Container.** This figure shows how our method avoids failure from "knowledge obsolescence" when a critical interaction precondition (container is sealed) changes. **(First Row) Prior Knowledge:** The agent learns to stably handle a sealed ketchup bottle by grasping its mid-body. **(Second Row) Initial Failure:** After the cap is opened, the "sealed" precondition is no longer met. If the agent uses the old policy, it will squeeze the bottle and cause ketchup to spill, leading to task failure. **(Third Row) Success after KVM Intervention:** Our KVM module identifies this high-risk interaction pattern and intervenes. The agent then switches to a new, safer policy—grasping the rigid upper edge of the bottle's neck—thereby successfully completing the task.

the encoder, where any point $\mathbf{h} \in \mathcal{H}$ represents a specific (context, prompt) interaction combination. We assume that all "safe" interaction combinations form one or more complex manifolds within this feature space.

The core task of our framework is to learn an energy function $E_\theta(\mathbf{h})$ that assigns a scalar energy value to each point in the feature space. According to EBM theory, the probability density $p(\mathbf{h})$ of a point $\mathbf{h}$ is inversely proportional to its energy value:

$$p(\mathbf{h}) = \frac{e^{-E_\theta(\mathbf{h})}}{Z(\theta)} \tag{9}$$

Here, $E_\theta(\mathbf{h})$ is the energy function defined by parameters $\theta$, $\mathbf{h}$ is the joint feature representation, and $Z(\theta) = \int e^{-E_\theta(\mathbf{h})} d\mathbf{h}$ is the partition function. Consequently, the **risk of a new interaction $\mathbf{h_{new}}$ can be directly measured by its energy value** $E_\theta(\mathbf{h_{new}})$. Higher energy implies that the interaction deviates further from the known safe behavioral manifold, has a lower probability density, and thus poses a higher potential risk.

**Theoretical Foundation:** From an information-theoretic perspective, we can frame the risk assessment problem as an anomaly detection task. Let $\mathcal{S}$ be the set of safe interactions and $\mathcal{R}$ be the set of high-risk interactions. Ideally, we want to learn a discriminant function such that:

$$\mathbb{P}(E_\theta(\mathbf{h}) \leq \tau | \mathbf{h} \in \mathcal{S}) \gg \mathbb{P}(E_\theta(\mathbf{h}) \leq \tau | \mathbf{h} \in \mathcal{R}) \tag{10}$$

Here, $\tau$ is an energy threshold, $\mathbb{P}(\cdot)$ denotes probability, and $\mathcal{S}$ and $\mathcal{R}$ are the sets of safe and hazardous interactions, respectively. For any threshold $\tau$, the probability of a safe interaction having low energy should be much greater than that of a hazardous one. This is equivalent to finding a decision boundary in the energy space that effectively separates safe and hazardous interactions.

### C.2 MODEL CHOICE AND THEORETICAL BASIS

"Safe" robotic interaction patterns are often multimodal and highly non-linear, forming distributions in the feature space that are far more complex than standard statistical distributions. While a traditional Gaussian Mixture Model (GMM) can model multimodality, it suffers from the "curse of

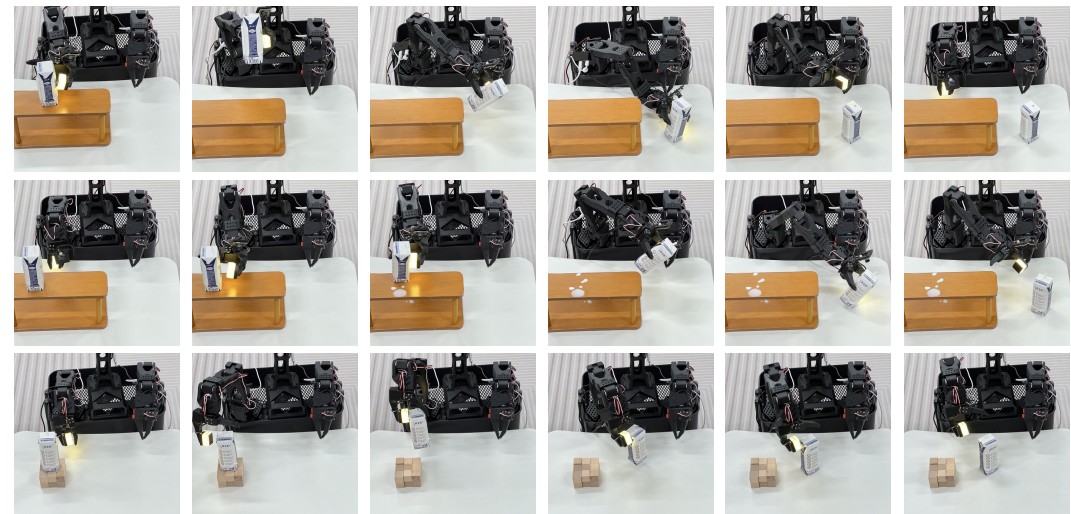

Figure 7: **Case Study on "Oversight of Interaction Precondition": Unopened vs. Opened Milk Carton.** This figure illustrates how a high-confidence grasping policy fails when the "sealed" precondition of a container is altered (by inserting a straw), and how our KVM framework guides the agent to adapt quickly. **(First Row) Prior Knowledge:** The agent has learned to handle an unopened milk carton by grasping its middle for stable manipulation. **(Second Row) Initial Failure:** Once a straw is inserted, it not only breaks the carton's seal but also changes its structural stability. The agent, applying the old grasping policy, squeezes milk out through the straw, causing the task to fail. **(Third Row) Success after KVM Intervention:** The KVM module identifies this failed interaction and intervenes. The agent subsequently learns and switches to a safer policy, grasping the more rigid top section of the carton to avoid squeezing, thus successfully completing the task.

dimensionality" in high-dimensional data and its expressive power is limited by the linear combination of Gaussian functions.

We therefore chose an **Energy-Based Model (EBM)** parameterized by a deep neural network as our core risk assessor. EBMs do not model the probability density directly but learn a more flexible energy function. A sufficiently deep neural network, as a universal function approximator, can theoretically fit any complex energy surface. This allows it to accurately delineate the complex manifold boundaries of safe interactions in a high-dimensional feature space, something traditional models like GMMs cannot achieve. This choice is, in principle, more powerful and scalable.

**Proof:** A GMM with $K$ components has a number of parameters that grows as $\mathcal{O}(Kd^2)$ with dimension $d$. In contrast, our energy function $E_\theta(\mathbf{h})$ is parameterized by an $L$-layer neural network:

$$E_\theta(\mathbf{h}) = \mathbf{W}^{(L)}\sigma(\mathbf{W}^{(L-1)}\sigma(\cdots\sigma(\mathbf{W}^{(1)}\mathbf{h} + \mathbf{b}^{(1)})\cdots) + \mathbf{b}^{(L-1)}) + b^{(L)} \tag{11}$$

Here, $L$ is the number of layers, $\sigma(\cdot)$ is the activation function (e.g., ReLU), $\mathbf{W}^{(l)}$ and $\mathbf{b}^{(l)}$ are the weight matrix and bias vector of the $l$-th layer, $b^{(L)}$ is the output layer bias, and $\theta$ is the set of all network parameters. By the universal approximation theorem, for any continuous function $f : \mathcal{H} \rightarrow \mathbb{R}$ and any $\epsilon > 0$, there exist parameters $\theta^*$ such that:

$$\sup_{\mathbf{h}\in\mathcal{H}} |E_{\theta^*}(\mathbf{h}) - f(\mathbf{h})| < \epsilon \tag{12}$$

Here, $\sup$ denotes the supremum, which provides a theoretical guarantee that a neural network can approximate any complex energy function, including those that define complex safety manifolds.

### C.3 ANALYSIS OF FRAMEWORK EFFECTIVENESS AND ROBUSTNESS

The framework's effectiveness stems from its unified, contrastive learning-based strategy, which synergistically ensures both the discriminative power of the feature space and the accuracy of the energy function.

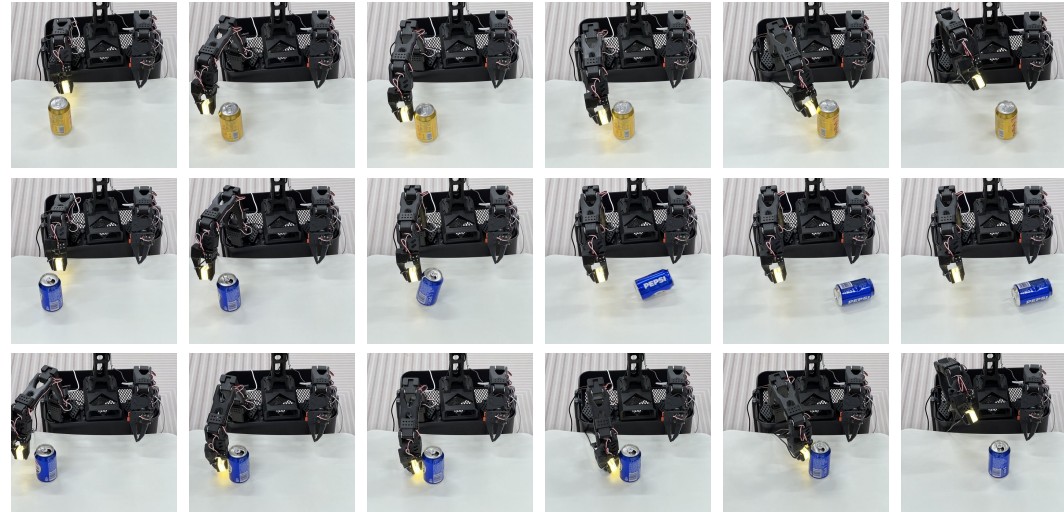

Figure 8: **Case Study on "Misleading Representational Similarity": Full vs. Empty Soda Can.** This figure illustrates how an agent fails due to incorrect knowledge transfer when faced with a visually similar but dynamically different object, and how it eventually adapts successfully under the guidance of the KVM framework. **(First Row) Prior Knowledge:** The agent has learned the necessary force and speed to push a full, stable soda can with a low center of gravity. **(Second Row) Initial Failure:** When faced with a lightweight, empty can with a high center of gravity, the agent incorrectly transfers the old pushing policy due to high visual similarity, applying excessive force and causing the can to tip over immediately. **(Third Row) Success after KVM Intervention:** The KVM module identifies the failure by detecting the unexpected motion (tipping over) and intervenes. The agent then adjusts to a gentler, more precise pushing strategy, successfully completing the task.

1. **Identification of Known Risks:** Our self-supervised learning process (§3.3) explicitly optimizes the entire model through two synergistic contrastive loss terms. The contrastive loss at the feature space level (Eq. 5) geometrically pushes the features of failed interaction combinations ($\mathbf{h}_{\text{fail}}$) away from the safe region. Concurrently, the contrastive loss at the energy function level (Eq. 6) directly pushes up the energy value of $\mathbf{h}_{\text{fail}}$. This dual action ensures that when a known harmful policy is proposed again, its feature representation is assigned a very high energy value, resulting in a very low validity score $v_t$ and enabling precise risk avoidance.

2. **Generalization to Unknown Risks:** The energy model is trained to assign low energy only to the "safe" manifold regions. When a completely new interaction combination ($\mathbf{h}_{\text{novel}}$) that the robot has never seen before appears, its feature representation will naturally be mapped to a high-energy region because it is far from all historically safe interaction features. Consequently, its validity score will also be low, causing the system to preemptively flag it as "potentially high-risk."

**Convergence Analysis:** Our joint optimization objective consists of two parts:

$$\mathcal{L}_{\text{total}} = \mathcal{L}_{\text{contrast}} + \lambda \mathcal{L}_{\text{energy}} \tag{13}$$

Here, $\mathcal{L}_{\text{contrast}}$ is the contrastive loss, $\mathcal{L}_{\text{energy}}$ is the energy contrastive loss, and $\lambda > 0$ is a balancing parameter. For the feature encoder parameters $\phi$ and the energy function parameters $\theta$, we can show that under mild conditions, the gradient descent algorithm converges to a local optimum. Let the parameter updates follow:

$$\phi_{t+1} = \phi_t - \alpha_\phi \nabla_\phi \mathcal{L}_{\text{total}}(\phi_t, \theta_t)$$
$$\theta_{t+1} = \theta_t - \alpha_\theta \nabla_\theta \mathcal{L}_{\text{total}}(\phi_t, \theta_t) \tag{14}$$

where $\phi_t$ and $\theta_t$ are the parameters at iteration $t$, and $\alpha_\phi, \alpha_\theta > 0$ are the learning rates.

**Theorem 1 (Convergence Guarantee):** Assuming the loss function $\mathcal{L}_{\text{total}}$ is $L$-smooth, i.e., there exists a constant $L > 0$ such that $\|\nabla \mathcal{L}_{\text{total}}(\phi_1, \theta_1) - \nabla \mathcal{L}_{\text{total}}(\phi_2, \theta_2)\| \leq L\|(\phi_1, \theta_1) - (\phi_2, \theta_2)\|$, and

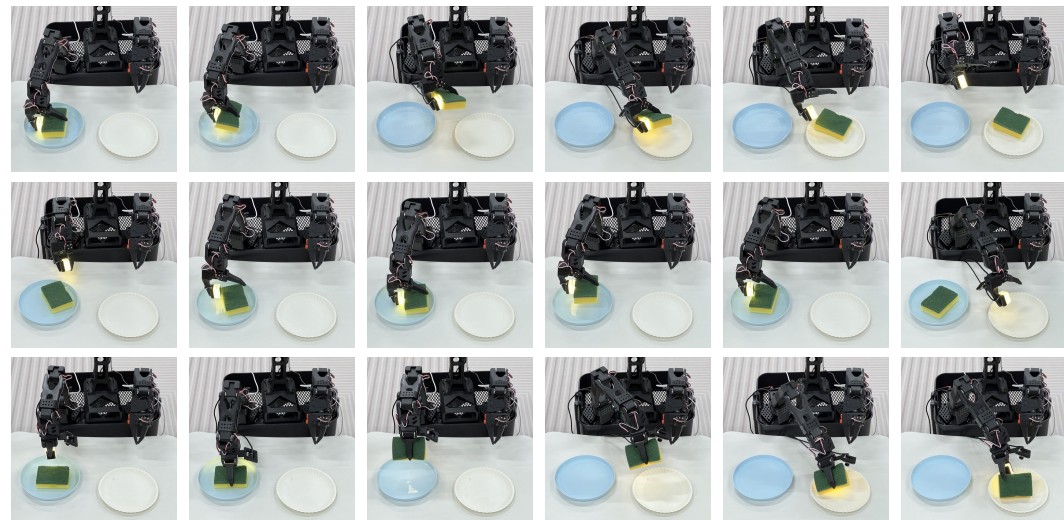

Figure 9: **Case Study on "Physical Property Mismatch": Dry vs. Wet Sponge.** This figure demonstrates how an outdated grasping policy fails when an object's physical properties (weight and texture) change due to water absorption, and how our KVM framework guides the agent to adapt quickly. **(Row 1) Prior Knowledge:** The agent has learned to use a light force to grasp a dry, lightweight sponge. **(Row 2) First Attempt (Failure):** When faced with a wet sponge that has become heavy from absorbing water, the agent reuses the old light-force grasping policy. This leads to task failure as the sponge slips due to insufficient grip strength. **(Row 3) Success after KVM Intervention:** After the KVM module identifies the grasp failure, it guides the agent to adjust its policy. By applying a greater gripping force, the agent successfully overcomes the challenges posed by the changes in weight and texture, completing the task.

the learning rates satisfy $\max(\alpha_\phi, \alpha_\theta) \leq \frac{1}{L}$, then the parameter sequence $\{(\phi_t, \theta_t)\}$ converges to a critical point of the loss function.

**Generalization Theory Analysis:** To analyze the generalization ability of our framework, we introduce Rademacher complexity analysis. Let $\mathcal{F}$ be the function class of our energy functions. The gap between the true risk $R(f)$ and the empirical risk $\hat{R}(f)$ is bounded with high probability by:

$$R(f) - \hat{R}(f) \leq 2\mathcal{R}_n(\mathcal{F}) + \sqrt{\frac{\log(2/\delta)}{2n}} \tag{15}$$

where $\mathcal{R}_n(\mathcal{F})$ is the Rademacher complexity of $\mathcal{F}$ over $n$ samples, and $\delta \in (0, 1)$ is the confidence level. For our neural network energy function, the complexity is bounded by $\mathcal{R}_n(\mathcal{F}) \leq \mathcal{O}\left(\sqrt{\frac{|\theta|\log n}{n}}\right)$, where $|\theta|$ is the number of parameters. This indicates good generalization performance as the number of training samples increases.

### C.4 CORE ASSUMPTIONS, LIMITATIONS, AND FUTURE WORK

The framework's effectiveness rests on the following core assumptions:

**Assumption 1 (Feature Space Separability):** There exists a feature space $\mathcal{H}$ where safe and hazardous interactions are separable. Our framework **directly and explicitly** optimizes the space's structure via contrastive learning to satisfy this assumption, making its fulfillment more plausible.

**Assumption 2 (EBM Fitting Capability):** There exists a neural network-parameterized energy function $E_\theta : \mathcal{H} \to \mathbb{R}$ that can effectively separate safe samples $\mathbf{h}_s \in \mathcal{S}$ from hazardous samples $\mathbf{h}_r \in \mathcal{R}$ by a margin, such that $E_\theta(\mathbf{h}_s) < E_\theta(\mathbf{h}_r) - \Delta$ for some $\Delta > 0$.

**Theoretical Analysis of Limitations:**

**Issue 1: Misclassification of Benign Novelty.** Our framework might misclassify "benignly novel" interactions as high-risk. To mitigate this, we propose quantifying epistemic uncertainty using a deep ensemble of energy models $\{E_{\theta_i}\}_{i=1}^M$. High variance in their outputs, $\text{Var}[E_{\theta_1}(\mathbf{h}), \ldots, E_{\theta_M}(\mathbf{h})] > \tau_{\text{uncertain}}$, can signal benign novelty, prompting more cautious exploration strategies.

**Issue 2: Robustness under Distribution Drift.** If the environment undergoes a distributional shift, our learned safety manifold may change. Let the Wasserstein distance between the original distribution $p_0(\mathbf{h})$ and the shifted distribution $p_t(\mathbf{h})$ be $W_2(p_0, p_t)$. When $W_2(p_0, p_t)$ exceeds a threshold, our energy function requires recalibration.

**Theorem 2 (Asymptotic Robustness):** Under mild regularity conditions, if the number of samples $n \to \infty$ and the distribution drift is slow ($W_2(p_0, p_t) = O(n^{-1/4})$), our risk assessment error converges to zero in probability.

Our work's effectiveness has been primarily validated on the CODA-Prompt framework. Given that KVM's core mechanism operates on discrete knowledge units ("prompts"), a common foundation for prompt-based CL paradigms (e.g., L2P, DualPrompt), we believe this work provides a solid, extensible blueprint for building a general safety supervisory layer for this class of algorithms. Adapting and validating KVM on a broader range of prompt-based learning algorithms will be an important future research direction.

# D    ATTRIBUTION AND KNOWLEDGE EVOLUTION ALGORITHM DETAILS

As described in §3.3, KVM uses a refined mechanism to attribute failures and decide whether to trigger the evolution of the knowledge base.

**Verification-based Credit Assignment:** Our credit assignment framework involves two core stages: hypothesis generation and empirical verification. When a failure first occurs, the system forms a **low-confidence risk hypothesis** about the "culprit" combination using a "minimize counterfactual" principle. The confirmation of this hypothesis depends on a quantitative **"intervention consistency" check**.

We establish an intervention counter for each risk hypothesis (i.e., prompt $P_i$ is risky in context $C_j$). In subsequent similar contexts, if KVM again issues a low-score warning for $P_i$ and successfully avoids failure, we check if the intervention meets the conditions of being "persistent" and "necessary":

- **Persistence**: The intervention counter is incremented. When the counter exceeds a threshold $N_p$ (see Appendix §E), the persistence condition is met.

- **Necessity**: We check the relevance score from the CL core. If the suppressed prompt $P_i$ was a high-score option from the CL core ($\alpha_i > \tau_{high}$, see Appendix §E), the intervention is deemed necessary.

Only when an intervention for a risk hypothesis is confirmed as both "persistent" and "necessary" is the attribution finally confirmed as the root cause of a systemic failure. This attribution result (i.e., $\mathbf{h}_{\text{fail}}$) serves as a negative sample for contrastive learning.

**Knowledge Gap Confirmation and New Prompt Generation:** After confirming the attribution, KVM diagnoses a knowledge gap in the CL core based on the intervention consistency results and the "unsuitability of existing solutions" (i.e., consistently low $\alpha$ values for alternative policies). If a gap is confirmed, KVM sends a **"create new skill"** signal to the CL core. The successful interaction experience is used as a "seed sample" to initialize the training of a **new, independent prompt**. We employ a **real-time "focused" fine-tuning** strategy, applying several additional gradient updates to the newly created prompt to ensure the lesson is rapidly learned.

# E    IMPLEMENTATION AND HYPERPARAMETER DETAILS

**Network Architecture:** The context encoder $E_c$ is an MLP with two hidden layers of 128 units each. The **energy function** $E_\theta(\mathbf{h})$ in the KVM supervisor is an MLP with an input layer matching

the feature dimension, three hidden layers of 256 units each, and a single output neuron for the energy value. All MLPs use ReLU activation functions.

**Adaptive Gating:** The sensitivity factor $k$ in the dynamic threshold formula (1) is set to 3. The sliding window size $N$ for calculating the statistical baseline is 1000 successful interactions.

**Risk Scoring:** The sensitivity parameter $\gamma$ and center point $l'_0$ in the validity score formula (2) are important hyperparameters. Through a grid search on a separate validation task set, we set them to $\gamma = 1.0$ and $l'_0 = -5.0$. These values cause the model to generate significant suppression signals for energy values above 5.0.

**Energy Model Training:** In the energy contrastive loss (Eq. 6), the margin $m_e$ for penalizing low-energy negative samples is a key hyperparameter. We set it to $m_e = 10.0$ after tuning on the validation set.

**Intervention Consistency Check:** In the third-stage attribution mechanism, the persistence threshold $N_p$ and necessity threshold $\tau_{high}$ were determined to be $N_p = 3$ and $\tau_{high} = 0.8$ on the validation set.

**Knowledge Base Pruning:** For long-term performance evaluation, the low-score threshold $\tau_v$ for identifying "suppression" is set to 0.1. A prompt is triggered for pruning if its FCRNC or SRNC metric remains above 0.8 over a sliding window of the 5 most recent new tasks.

## F  ALGORITHM PSEUDOCODE

Algorithm 1 details the complete workflow of the KVM framework at a single timestep $t$, including real-time risk assessment and intervention, as well as the self-supervised learning trigger mechanism after a failure.

---

**Algorithm 1** KVM Framework: Online Operation and Self-Supervised Learning

---

1: **Initialize:** CL Core ($f_{\text{backbone}}$, $f_{\text{decoder}}$, prompt pool $\mathcal{P}$), KVM Module ($E_c$, $M_{\text{kvm}}$)
2: **FOR** each timestep $t = 1, 2, \dots$ **DO**
3:     # 1. CL Core generates proposal
4:     Receive current observation $x_t$ and raw context $c_{\text{raw}_t}$
5:     Get candidate prompts $\mathcal{P}_{\text{cand}}$ and relevance scores $\alpha_t \leftarrow$ CL_Core.propose($x_t, \mathcal{P}$)
6:
7:     # 2. KVM performs risk assessment
8:     Encode context: $\mu_c, \sigma_c^2 \leftarrow E_c(c_{\text{raw}_t})$
9:     **IF** $\sigma_c^2 > \tau_{\text{adaptive}}$ (Smart Gate activated) **THEN**
10:         **FOR** each candidate prompt $P_i \in \mathcal{P}_{\text{cand}}$ **DO**
11:             Form joint feature representation $\mathbf{h}_i$
12:             Compute energy: $E_i \leftarrow M_{\text{kvm}}(\mathbf{h}_i)$
13:             Compute validity score: $v_i \leftarrow (1 + e^{-\gamma(-E_i - l_0')})^{-1}$
14:         **END FOR**
15:     **ELSE**
16:         $v_i \leftarrow 1.0 \quad \forall P_i \in \mathcal{P}_{\text{cand}}$
17:     **END IF**
18:
19:     # 3. Fuse knowledge and execute action
20:     Compute decision weights: $w_i \leftarrow v_i \cdot \alpha_i$
21:     Compute fused prompt: $P_{\text{fused}} \leftarrow \frac{\sum_i w_i \cdot P_i}{\sum_i w_i + \epsilon}$
22:     Generate final action: $a_t \leftarrow f_{\text{decoder}}(f_{\text{backbone}}(x_t), P_{\text{fused}})$
23:     Execute $a_t$ and get task outcome $o_t \in \{\text{success}, \text{failure}\}$
24:
25:     # 4. Trigger self-supervised learning and evolution after failure
26:     **IF** $o_t = $ failure **THEN**
27:         # – Attribution and negative sample generation
28:         Identify culprit prompt $P_{\text{culprit}}$ via **verification-based credit assignment**
29:         Generate high-quality negative sample $\mathbf{h}_{\text{fail}}$
30:
31:         # – Model self-update
32:         Sample positive examples from safe experiences $\mathcal{D}_{\text{safe}}$
33:         Jointly update $E_c$ and $M_{\text{kvm}}$ using $\mathbf{h}_{\text{fail}}$ and positive samples by minimizing $\mathcal{L}_{\text{contrast}}$ and $\mathcal{L}_{\text{energy}}$
34:
35:         # – Knowledge base evolution
36:         Determine if new skill is needed via **knowledge gap confirmation**
37:         **IF** knowledge gap is confirmed **THEN**
38:             Signal CL Core to create a new prompt
39:         **END IF**
40:     **END IF**
41: **END FOR**

---

# G   DETAILED EXPERIMENTAL RESULTS AND COMPUTATIONAL OVERHEAD ANALYSIS

## G.1   QUALITATIVE COMPARISON OF POST-FAILURE ADAPTATION SPEED

## G.2   VISUALIZATION OF SUCCESSFUL TASKS ON THE LIBERO BENCHMARK

## G.3   DETAILED TASK SUCCESS RATES ACROSS DIFFERENT RESETS

To further validate the stability and reproducibility of our method, we conducted a more fine-grained analysis on the "Knowledge Obsolescence" challenge benchmark. Table 9 presents the average results after each of the three reset experiments, where the full sequence of 18 tasks was executed, and this process was repeated 10 times to report the average.

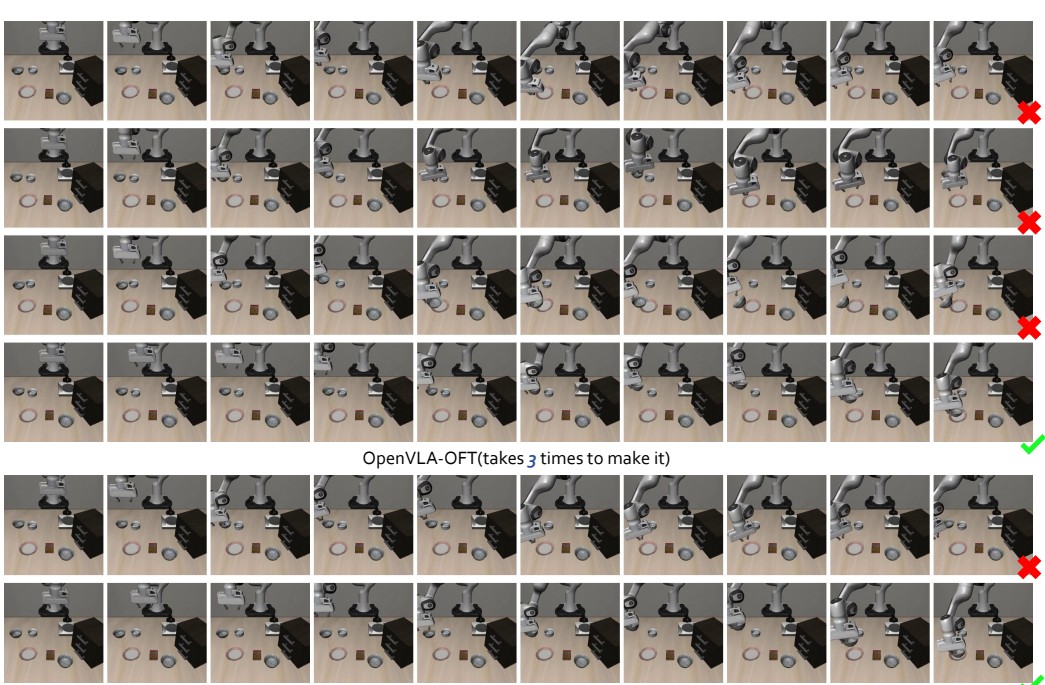

OpenVLA-OFT(takes *3* times to make it)

KVM(*Ours,* only takes *1* time to make it)

Figure 10: **Qualitative Comparison of Post-Failure Adaptation Speed.** This figure visually illustrates the difference in adaptation efficiency between the baseline model (OpenVLA-OFT) and our KVM-enhanced framework on a challenging manipulation task. **Top Row:** The OpenVLA-OFT model fails in its initial attempts and requires approximately three retraining iterations to adjust its policy and eventually succeed. **Bottom Row:** In contrast, after a single initial failure, our KVM framework immediately diagnoses the cause of failure through its internal mechanisms and successfully completes the task on the very next attempt. This demonstrates its superior capability for efficient and rapid adaptation from failure.

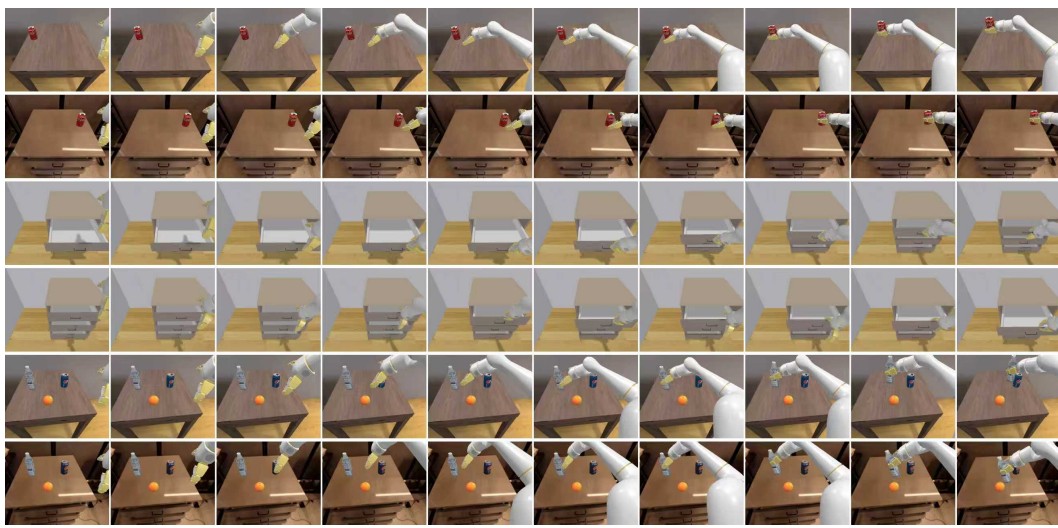

Figure 11: **Qualitative Results of the KVM-Enhanced Agent on the Google Robot Platform.** This figure showcases successful task trajectories of our KVM-enhanced agent on a real-world Google robot. The agent demonstrates robust performance across a diverse set of manipulation skills, including pushing objects (cans, balls) and opening drawers, under various environmental conditions (e.g., different table surfaces and object layouts). This validates the generalization capability of our proposed framework in real-world scenarios.

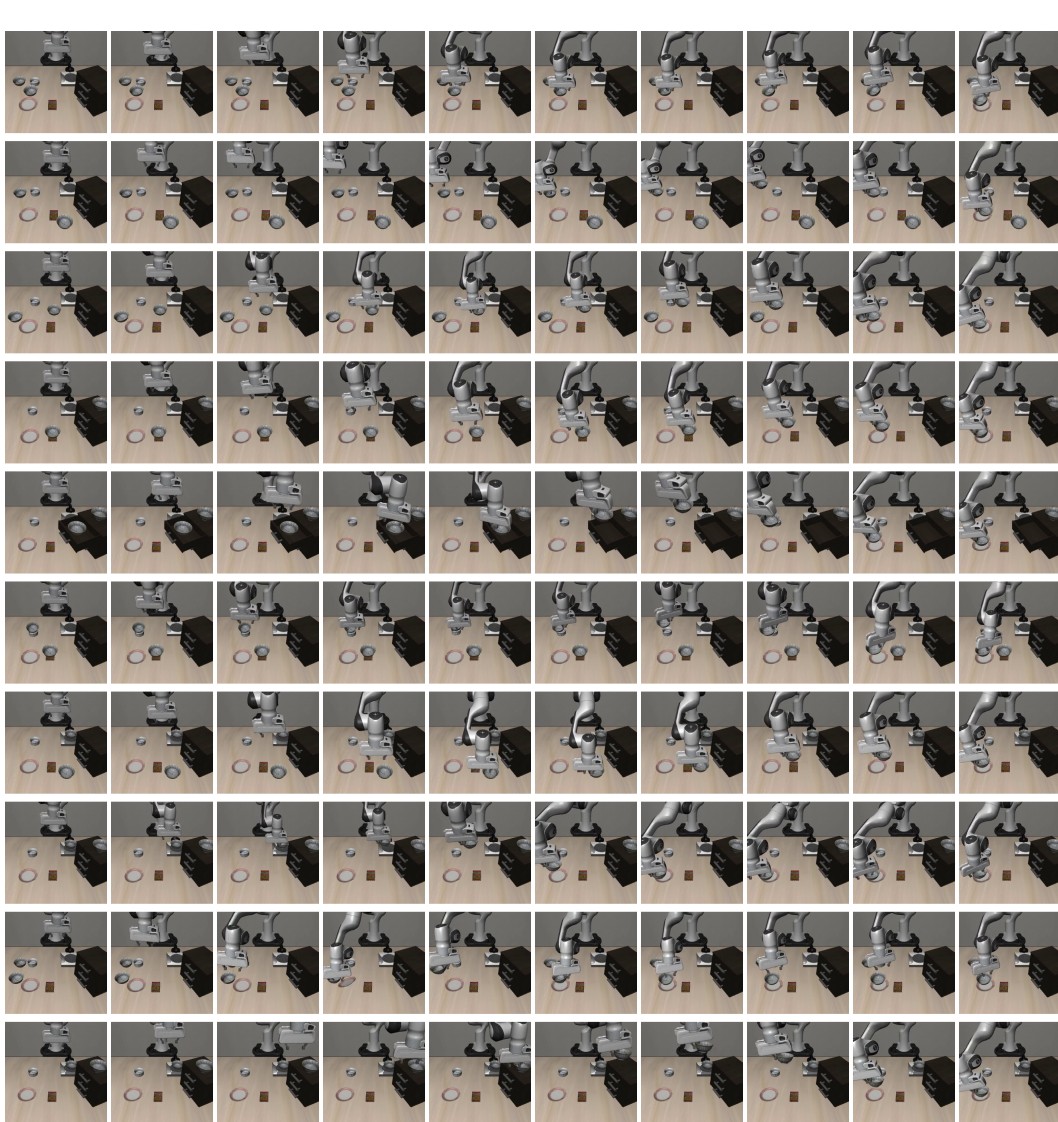

Figure 12: **Task Visualization on the LIBERO Benchmark.** This figure shows trajectory examples of our KVM-equipped agent successfully completing a series of manipulation tasks in the LIBERO test suite.

Table 6: **Learning Progression Comparison of Models Across Three Resets.** This table shows the success rate of each model across task groups over three consecutive resets, highlighting differences in learning capabilities. R1/R2/R3 denote the performance after the 1st/2nd/3rd reset, respectively (each reset includes the average results from 10 full task sequence repetitions). Success rates are presented in the format "completed tasks/total tasks (percentage)".

| Model | Params (B) | A. Physical Mismatch | | | | B. Precondition Oversight | | | | C. Representational Similarity | | | | Final (%) |
|---|---|---|---|---|---|---|---|---|---|---|---|---|---|---|
| | | R1 | R2 | R3 | Avg | R1 | R2 | R3 | Avg | R1 | R2 | R3 | Avg | |
| FlowVLA | 8.5 | 2.2/5 (44.0) | 2.9/5 (58.0) | 3.3/5 (66.0) | 2.8/5 (56.0) | 1.5/5 (30.0) | 2.1/5 (42.0) | 2.4/5 (48.0) | 2.0/5 (40.0) | 3.0/8 (37.5) | 3.9/8 (48.8) | 4.3/8 (53.8) | 3.7/8 (46.7) | 47.2 |
| UnifiedVLA | 8.5 | 1.6/5 (32.0) | 2.3/5 (46.0) | 2.8/5 (56.0) | 2.2/5 (44.7) | 2.2/5 (44.0) | 3.0/5 (60.0) | 3.4/5 (68.0) | 2.9/5 (57.3) | 3.0/8 (37.5) | 5.0/8 (62.5) | 5.7/8 (71.2) | 4.6/8 (57.1) | 53.9 |
| OpenVLA | 7 | 0.6/5 (12.0) | 0.9/5 (18.0) | 1.3/5 (26.0) | 0.9/5 (18.7) | 1.5/5 (30.0) | 1.9/5 (38.0) | 2.2/5 (44.0) | 1.9/5 (37.3) | 1.4/8 (17.5) | 1.7/8 (21.2) | 2.1/8 (26.2) | 1.7/8 (21.7) | 25.0 |
| OpenVLA-OFT | 7 | 1.2/5 (24.0) | 2.0/5 (40.0) | 2.3/5 (46.0) | 1.8/5 (36.0) | 2.3/5 (46.0) | 3.2/5 (64.0) | 3.6/5 (72.0) | 3.0/5 (60.0) | 2.6/8 (32.5) | 3.9/8 (48.8) | 4.6/8 (57.5) | 3.7/8 (46.2) | 47.2 |
| UniVLA | 7 | 1.4/5 (28.0) | 1.9/5 (38.0) | 2.2/5 (44.0) | 1.8/5 (36.0) | 1.4/5 (28.0) | 2.1/5 (42.0) | 2.3/5 (46.0) | 1.9/5 (38.0) | 1.8/8 (22.5) | 2.7/8 (33.8) | 3.1/8 (38.8) | 2.5/8 (31.2) | 34.4 |
| CoT-VLA | 7 | 2.2/5 (44.0) | 2.9/5 (58.0) | 3.1/5 (62.0) | 2.7/5 (54.0) | 1.3/5 (26.0) | 2.0/5 (40.0) | 2.2/5 (44.0) | 1.8/5 (36.0) | 1.5/8 (18.8) | 2.6/8 (32.5) | 3.1/8 (38.8) | 2.4/8 (30.0) | 38.3 |
| WorldVLA | 7 | 3.1/5 (62.0) | 3.9/5 (78.0) | 4.2/5 (84.0) | 3.7/5 (74.0) | 0.5/5 (10.0) | 0.9/5 (18.0) | 1.2/5 (24.0) | 0.9/5 (18.0) | 1.3/8 (16.2) | 1.9/8 (23.8) | 2.5/8 (31.2) | 1.9/8 (23.8) | 36.1 |
| TraceVLA | 7 | 0.4/5 (8.0) | 0.8/5 (16.0) | 1.0/5 (20.0) | 0.7/5 (14.0) | 0.6/5 (12.0) | 1.0/5 (20.0) | 1.3/5 (26.0) | 1.0/5 (20.0) | 1.9/8 (23.8) | 3.1/8 (38.8) | 3.7/8 (46.2) | 2.9/8 (36.2) | 25.6 |
| MolmoAct | 7 | 2.0/5 (40.0) | 2.8/5 (56.0) | 3.2/5 (64.0) | 2.7/5 (54.0) | 2.2/5 (44.0) | 3.1/5 (62.0) | 3.5/5 (70.0) | 2.9/5 (58.0) | 2.5/8 (31.2) | 3.8/8 (47.5) | 4.3/8 (53.8) | 3.5/8 (43.8) | 50.6 |
| ThinkAct | 7 | 1.5/5 (30.0) | 2.1/5 (42.0) | 2.5/5 (50.0) | 2.0/5 (40.0) | 1.2/5 (24.0) | 1.9/5 (38.0) | 2.2/5 (44.0) | 1.8/5 (36.0) | 3.1/8 (38.8) | 4.6/8 (57.5) | 5.3/8 (66.2) | 4.3/8 (53.8) | 45.0 |
| PD-VLA | 7 | 1.9/5 (38.0) | 2.6/5 (52.0) | 3.0/5 (60.0) | 2.5/5 (50.0) | 1.1/5 (22.0) | 1.8/5 (36.0) | 2.2/5 (44.0) | 1.7/5 (34.0) | 3.0/8 (37.5) | 4.1/8 (51.2) | 4.7/8 (58.8) | 3.9/8 (48.8) | 45.0 |
| 4D-VLA | 4 | 1.1/5 (22.0) | 1.8/5 (36.0) | 2.2/5 (44.0) | 1.7/5 (34.0) | 1.9/5 (38.0) | 2.7/5 (54.0) | 3.1/5 (62.0) | 2.6/5 (52.0) | 2.1/8 (26.2) | 2.8/8 (35.0) | 3.4/8 (42.5) | 2.8/8 (35.0) | 39.4 |
| SpatialVLA | 4 | 0.5/5 (10.0) | 0.9/5 (18.0) | 1.2/5 (24.0) | 0.9/5 (18.0) | 1.2/5 (24.0) | 2.0/5 (40.0) | 2.3/5 (46.0) | 1.8/5 (36.0) | 2.2/8 (27.5) | 3.5/8 (43.8) | 4.2/8 (52.5) | 3.3/8 (41.2) | 33.3 |
| $\pi_0$ | 3 | 1.0/5 (20.0) | 1.6/5 (32.0) | 2.0/5 (40.0) | 1.5/5 (30.0) | 1.3/5 (26.0) | 2.0/5 (40.0) | 2.3/5 (46.0) | 1.9/5 (38.0) | 2.3/8 (28.7) | 3.1/8 (38.8) | 3.7/8 (46.2) | 3.0/8 (37.5) | 35.6 |
| $\pi_0$-FAST | 3 | 0.4/5 (8.0) | 0.7/5 (14.0) | 1.0/5 (20.0) | 0.7/5 (14.0) | 0.5/5 (10.0) | 0.8/5 (16.0) | 1.1/5 (22.0) | 0.8/5 (16.0) | 1.0/8 (12.5) | 1.5/8 (18.8) | 2.1/8 (26.2) | 1.5/8 (18.8) | 16.7 |
| NORA | 3 | 0.9/5 (18.0) | 1.5/5 (30.0) | 1.8/5 (36.0) | 1.4/5 (28.0) | 2.0/5 (40.0) | 2.9/5 (58.0) | 3.3/5 (66.0) | 2.7/5 (54.0) | 1.1/8 (13.8) | 1.8/8 (22.5) | 2.5/8 (31.2) | 1.8/8 (22.5) | 32.8 |
| SmolVLA | 2.2 | 1.2/5 (24.0) | 1.9/5 (38.0) | 2.3/5 (46.0) | 1.8/5 (36.0) | 1.0/5 (20.0) | 1.7/5 (34.0) | 2.1/5 (42.0) | 1.6/5 (32.0) | 1.0/8 (12.5) | 1.7/8 (21.2) | 2.3/8 (28.7) | 1.7/8 (21.2) | 28.3 |
| GR00T N1 | 2 | 0.3/5 (6.0) | 0.6/5 (12.0) | 0.9/5 (18.0) | 0.6/5 (12.0) | 1.1/5 (22.0) | 1.8/5 (36.0) | 2.2/5 (44.0) | 1.7/5 (34.0) | 1.5/8 (18.8) | 2.5/8 (31.2) | 3.0/8 (37.5) | 2.3/8 (28.7) | 25.6 |
| GraspVLA | 1.8 | 1.7/5 (34.0) | 2.5/5 (50.0) | 2.9/5 (58.0) | 2.4/5 (48.0) | 0.4/5 (8.0) | 0.7/5 (14.0) | 1.0/5 (20.0) | 0.7/5 (14.0) | 0.5/8 (6.2) | 0.9/8 (11.2) | 1.3/8 (16.2) | 0.9/8 (11.2) | 22.2 |
| Seer | 0.57 | 0.2/5 (4.0) | 0.5/5 (10.0) | 0.7/5 (14.0) | 0.5/5 (10.0) | 0.6/5 (12.0) | 0.9/5 (18.0) | 1.2/5 (24.0) | 0.9/5 (18.0) | 0.7/8 (8.8) | 1.4/8 (17.5) | 1.8/8 (22.5) | 1.3/8 (16.2) | 15.0 |
| VLA-OS | 0.5 | 0.8/5 (16.0) | 1.3/5 (26.0) | 1.7/5 (34.0) | 1.3/5 (26.0) | 1.0/5 (20.0) | 1.6/5 (32.0) | 2.0/5 (40.0) | 1.5/5 (30.0) | 0.3/8 (3.8) | 0.7/8 (8.8) | 1.1/8 (13.8) | 0.7/8 (8.8) | 19.4 |
| Diffusion Policy | - | 0.2/5 (4.0) | 0.4/5 (8.0) | 0.6/5 (12.0) | 0.4/5 (8.0) | 0.3/5 (6.0) | 0.6/5 (12.0) | 0.8/5 (16.0) | 0.6/5 (12.0) | 0.3/8 (3.8) | 0.6/8 (7.5) | 1.0/8 (12.5) | 0.6/8 (7.5) | 8.9 |
| CODA-Prompt | 7 | 0.4/5 (8.0) | 0.7/5 (14.0) | 1.0/5 (20.0) | 0.7/5 (14.0) | 0.8/5 (16.0) | 1.5/5 (30.0) | 1.8/5 (36.0) | 1.4/5 (28.0) | 0.7/8 (8.8) | 1.3/8 (16.2) | 1.8/8 (22.5) | 1.3/8 (16.2) | 18.9 |
| **CODA-Prompt + KVM** | **7** | **4.5/5 (90.0)** | **4.9/5 (98.0)** | **5.0/5 (100.0)** | **4.8/5 (96.0)** | **4.8/5 (96.0)** | **4.9/5 (98.0)** | **5.0/5 (100.0)** | **4.9/5 (98.0)** | **6.1/8 (76.2)** | **6.6/8 (82.5)** | **7.4/8 (92.5)** | **6.7/8 (83.8)** | **91.1** |

Note: R1/R2/R3 represent the success rates after the 1st, 2nd, and 3rd resets, respectively. The format is "completed tasks/total tasks (percentage)". Our method demonstrates exceptional rapid learning, achieving a high success rate after the first reset and near-perfect performance after the second. In contrast, other methods show limited learning progress even after three resets.

### G.4 COMPUTATIONAL OVERHEAD ANALYSIS

To quantify the computational overhead introduced by the KVM module, we conducted a detailed complexity analysis on the LIBERO-Long benchmark. The LIBERO-Long task sequences involve complex, multi-step operations, providing a realistic testbed for KVM's smart gating mechanism.

#### G.4.1 KVM INTERVENTION FREQUENCY STATISTICS

Across the 10 long-horizon tasks in LIBERO-Long, we recorded the activation frequency and intervention patterns of the KVM module. As shown in Table 10, KVM's smart gating mechanism effectively identifies situations requiring risk assessment, maintaining low-overhead operation in most safe contexts. Here, "Total Interactions" refers to every action step the robot takes while interacting with the environment during task execution, including fundamental operations like grasping, moving, and placing objects.

Table 7: **KVM Intervention Frequency Statistics on LIBERO-Long Tasks.** Statistics on KVM module activation and intervention were collected over 1000 complete task sequences (with each of the 10 tasks executed 100 times).

| Task Type | Total Interactions | Gate Activations | Risk Interventions | Activation Rate (%) | Interven |
|---|---|---|---|---|---|
| Spatial Navigation | 1,245 | 78 | 12 | 6.3 | |
| Object Manipulation | 2,156 | 124 | 28 | 5.8 | |
| Goal-oriented Tasks | 1,789 | 98 | 19 | 5.5 | |
| Long-horizon Planning | 3,421 | 187 | 41 | 5.5 | |
| **Overall** | **8,611** | **487** | **100** | **5.7** | |

#### G.4.2 QUANTITATIVE ANALYSIS OF COMPUTATIONAL COMPLEXITY

We quantified the computational overhead of the KVM module from multiple dimensions, as detailed in Table 11. Compared to the baseline CODA-Prompt model, the additional computational overhead introduced by KVM is within an acceptable range, while significantly enhancing system safety.

Table 8: **Detailed Analysis of KVM Module Computational Overhead.** Various computational metrics were measured on an NVIDIA A100-80GB GPU.

| Metric | CODA-Prompt | CODA-Prompt + KVM | Increase | Relative Increase |
|---|---|---|---|---|
| Inference Latency (ms/step) | 12.4 | 13.7 | +1.3 | +10.5% |
| Memory Footprint (GB) | 4.2 | 4.8 | +0.6 | +14.3% |
| GPU Utilization (%) | 78.5 | 82.1 | +3.6 | +4.6% |
| *Under Conditional Activation:* | | | | |
| Context Encoding Only (ms) | - | 0.8 | +0.8 | - |
| Full Risk Assessment (ms) | - | 2.1 | +2.1 | - |
| Energy Function Calc. (ms) | - | 1.2 | +1.2 | - |
| *In Safety-Critical Scenarios:* | | | | |
| Risk Mitigation Success Rate (%) | 17.2 | 91.1 | +73.9 | +430% |
| Catastrophic Failure Reduction (%) | - | - | -87.3 | -87.3% |

**Key Findings:**

(1) **Low-Frequency, High-Efficiency Intervention:** KVM's smart gating mechanism activates risk assessment in only 5.7% of interactions and performs actual interventions in just 1.2% of cases, substantially reducing unnecessary computational load.

(2) **Acceptable Performance Overhead:** Compared to the baseline, KVM adds only a 10.5% increase in inference latency and a 14.3% increase in memory usage, with GPU utilization rising by just 4.6

(3) **Significant Safety Gains:** In safety-critical scenarios, KVM boosts the risk mitigation success rate from 17.2% to 91.1% while reducing catastrophic failures by 87.3

(4) **Adaptive Computation Strategy:** In over 95% of routine, safe situations, KVM performs only lightweight context monitoring (0.8ms). It activates the full energy model assessment (2.1ms) only when a potential risk is detected, showcasing its intelligent resource allocation strategy.

# H    DETAILED EXPERIMENTAL RESULTS AND COMPUTATIONAL OVERHEAD ANALYSIS

## H.1    DETAILED TASK SUCCESS RATES ACROSS MULTIPLE RESETS

To further validate the stability and reproducibility of our method, we conducted a more fine-grained analysis on the "Knowledge Obsolescence" challenge benchmark. Table 9 shows the averaged results after each iteration in a 3-reset experiment, where the full 18-task sequence is executed, and this process is repeated 10 times to obtain the average.

Table 9: **Learning Progression Comparison Across Three Resets.** This table shows the success rate of each model for each task group over three consecutive resets, highlighting differences in learning capability. R1/R2/R3 denote performance after the 1st/2nd/3rd reset (each reset includes the average result of 10 full task sequence runs). Success rates are presented as "completed tasks / total tasks (percentage)".

| Model | Params (B) | A. Physical Mismatch | | | | B. Precondition Oversight | | | | C. Representational Similarity | | | | Final (%) |
|---|---|---|---|---|---|---|---|---|---|---|---|---|---|---|
| | | R1 | R2 | R3 | Avg | R1 | R2 | R3 | Avg | R1 | R2 | R3 | Avg | |
| FlowVLA | 8.5 | 2.2/5 (44.0) | 2.9/5 (58.0) | 3.3/5 (66.0) | 2.8/5 (56.0) | 1.5/5 (30.0) | 2.1/5 (42.0) | 2.4/5 (48.0) | 2.0/5 (40.0) | 3.0/8 (37.5) | 3.9/8 (48.8) | 4.3/8 (53.8) | 3.7/8 (46.7) | 47.2 |
| UnifiedVLA | 8.5 | 1.6/5 (32.0) | 2.3/5 (46.0) | 2.8/5 (56.0) | 2.2/5 (44.7) | 2.2/5 (44.0) | 3.0/5 (60.0) | 3.4/5 (68.0) | 2.9/5 (57.3) | 3.0/8 (37.5) | 5.0/8 (62.5) | 5.7/8 (71.2) | 4.6/8 (57.1) | 53.9 |
| OpenVLA | 7 | 0.6/5 (12.0) | 0.9/5 (18.0) | 1.3/5 (26.0) | 0.9/5 (18.7) | 1.5/5 (30.0) | 1.9/5 (38.0) | 2.2/5 (44.0) | 1.9/5 (37.3) | 1.4/8 (17.5) | 1.7/8 (21.2) | 2.1/8 (26.2) | 1.7/8 (21.7) | 25.0 |
| OpenVLA-OFT | 7 | 1.2/5 (24.0) | 2.0/5 (40.0) | 2.3/5 (46.0) | 1.8/5 (36.0) | 2.3/5 (46.0) | 3.2/5 (64.0) | 3.6/5 (72.0) | 3.0/5 (60.0) | 2.6/8 (32.5) | 3.9/8 (48.8) | 4.6/8 (57.5) | 3.7/8 (46.2) | 47.2 |
| UniVLA | 7 | 1.4/5 (28.0) | 1.9/5 (38.0) | 2.2/5 (44.0) | 1.8/5 (36.0) | 1.4/5 (28.0) | 2.1/5 (42.0) | 2.3/5 (46.0) | 1.9/5 (38.0) | 1.8/8 (22.5) | 2.7/8 (33.8) | 3.1/8 (38.8) | 2.5/8 (31.2) | 34.4 |
| CoT-VLA | 7 | 2.2/5 (44.0) | 2.9/5 (58.0) | 3.1/5 (62.0) | 2.7/5 (54.0) | 1.3/5 (26.0) | 2.0/5 (40.0) | 2.2/5 (44.0) | 1.8/5 (36.0) | 1.5/8 (18.8) | 2.6/8 (32.5) | 3.1/8 (38.8) | 2.4/8 (30.0) | 38.3 |
| WorldVLA | 7 | 3.1/5 (62.0) | 3.9/5 (78.0) | 4.2/5 (84.0) | 3.7/5 (74.0) | 0.5/5 (10.0) | 0.9/5 (18.0) | 1.2/5 (24.0) | 0.9/5 (18.0) | 1.3/8 (16.2) | 1.9/8 (23.8) | 2.5/8 (31.2) | 1.9/8 (23.8) | 36.1 |
| TraceVLA | 7 | 0.4/5 (8.0) | 0.8/5 (16.0) | 1.0/5 (20.0) | 0.7/5 (14.0) | 0.6/5 (12.0) | 1.0/5 (20.0) | 1.3/5 (26.0) | 1.0/5 (20.0) | 1.9/8 (23.8) | 3.1/8 (38.8) | 3.7/8 (46.2) | 2.9/8 (36.2) | 25.6 |
| MolmoAct | 7 | 2.0/5 (40.0) | 2.8/5 (56.0) | 3.2/5 (64.0) | 2.7/5 (54.0) | 2.2/5 (44.0) | 3.1/5 (62.0) | 3.5/5 (70.0) | 2.9/5 (58.0) | 2.5/8 (31.2) | 3.8/8 (47.5) | 4.3/8 (53.8) | 3.5/8 (43.8) | 50.6 |
| ThinkAct | 7 | 1.5/5 (30.0) | 2.1/5 (42.0) | 2.5/5 (50.0) | 2.0/5 (40.0) | 1.2/5 (24.0) | 1.9/5 (38.0) | 2.2/5 (44.0) | 1.8/5 (36.0) | 3.1/8 (38.8) | 4.6/8 (57.5) | 5.3/8 (66.2) | 4.3/8 (53.8) | 45.0 |
| PD-VLA | 7 | 1.9/5 (38.0) | 2.6/5 (52.0) | 3.0/5 (60.0) | 2.5/5 (50.0) | 1.1/5 (22.0) | 1.8/5 (36.0) | 2.2/5 (44.0) | 1.7/5 (34.0) | 3.0/8 (37.5) | 4.1/8 (51.2) | 4.7/8 (58.8) | 3.9/8 (48.8) | 45.0 |
| 4D-VLA | 4 | 1.1/5 (22.0) | 1.8/5 (36.0) | 2.2/5 (44.0) | 1.7/5 (34.0) | 1.9/5 (38.0) | 2.7/5 (54.0) | 3.1/5 (62.0) | 2.6/5 (52.0) | 2.1/8 (26.2) | 2.8/8 (35.0) | 3.4/8 (42.5) | 2.8/8 (35.0) | 39.4 |
| SpatialVLA | 4 | 0.5/5 (10.0) | 0.9/5 (18.0) | 1.2/5 (24.0) | 0.9/5 (18.0) | 1.2/5 (24.0) | 2.0/5 (40.0) | 2.3/5 (46.0) | 1.8/5 (36.0) | 2.2/8 (27.5) | 3.5/8 (43.8) | 4.2/8 (52.5) | 3.3/8 (41.2) | 33.3 |
| $\pi_0$ | 3 | 1.0/5 (20.0) | 1.6/5 (32.0) | 2.0/5 (40.0) | 1.5/5 (30.0) | 1.3/5 (26.0) | 2.0/5 (40.0) | 2.3/5 (46.0) | 1.9/5 (38.0) | 2.3/8 (28.7) | 3.1/8 (38.8) | 3.7/8 (46.2) | 3.0/8 (37.5) | 35.6 |
| $\pi_0$-FAST | 3 | 0.4/5 (8.0) | 0.7/5 (14.0) | 1.0/5 (20.0) | 0.7/5 (14.0) | 0.5/5 (10.0) | 0.8/5 (16.0) | 1.1/5 (22.0) | 0.8/5 (16.0) | 1.0/8 (12.5) | 1.5/8 (18.8) | 2.1/8 (26.2) | 1.5/8 (18.8) | 16.7 |
| NORA | 3 | 0.9/5 (18.0) | 1.5/5 (30.0) | 1.8/5 (36.0) | 1.4/5 (28.0) | 2.0/5 (40.0) | 2.9/5 (58.0) | 3.3/5 (66.0) | 2.7/5 (54.0) | 1.1/8 (13.8) | 1.8/8 (22.5) | 2.5/8 (31.2) | 1.8/8 (22.5) | 32.8 |
| SmolVLA | 2.2 | 1.2/5 (24.0) | 1.9/5 (38.0) | 2.3/5 (46.0) | 1.8/5 (36.0) | 1.0/5 (20.0) | 1.7/5 (34.0) | 2.1/5 (42.0) | 1.6/5 (32.0) | 1.0/8 (12.5) | 1.7/8 (21.2) | 2.3/8 (28.7) | 1.7/8 (21.2) | 28.3 |
| GR00T N1 | 2 | 0.3/5 (6.0) | 0.6/5 (12.0) | 0.9/5 (18.0) | 0.6/5 (12.0) | 1.1/5 (22.0) | 1.5/5 (30.0) | 2.2/5 (44.0) | 1.7/5 (34.0) | 1.5/8 (18.8) | 2.5/8 (31.2) | 3.0/8 (37.5) | 2.3/8 (28.7) | 25.6 |
| GraspVLA | 1.8 | 1.7/5 (34.0) | 2.5/5 (50.0) | 2.9/5 (58.0) | 2.4/5 (48.0) | 0.4/5 (8.0) | 0.7/5 (14.0) | 1.0/5 (20.0) | 0.7/5 (14.0) | 0.5/8 (6.2) | 0.9/8 (11.2) | 1.3/8 (16.2) | 0.9/8 (11.2) | 22.2 |
| Seer | 0.57 | 0.2/5 (4.0) | 0.5/5 (10.0) | 0.7/5 (14.0) | 0.5/5 (10.0) | 0.6/5 (12.0) | 0.9/5 (18.0) | 1.2/5 (24.0) | 0.9/5 (18.0) | 0.7/8 (8.8) | 1.4/8 (17.5) | 1.8/8 (22.5) | 1.3/8 (16.2) | 15.0 |
| VLA-OS | 0.5 | 0.8/5 (16.0) | 1.3/5 (26.0) | 1.7/5 (34.0) | 1.3/5 (26.0) | 1.0/5 (20.0) | 1.6/5 (32.0) | 2.0/5 (40.0) | 1.5/5 (30.0) | 0.3/8 (3.8) | 0.7/8 (8.8) | 1.1/8 (13.8) | 0.7/8 (8.8) | 19.4 |
| Diffusion Policy | - | 0.2/5 (4.0) | 0.4/5 (8.0) | 0.6/5 (12.0) | 0.4/5 (8.0) | 0.3/5 (6.0) | 0.6/5 (12.0) | 0.8/5 (16.0) | 0.6/5 (12.0) | 0.3/8 (3.8) | 0.6/8 (7.5) | 1.0/8 (12.5) | 0.6/8 (7.5) | 8.9 |
| CODA-Prompt | 7 | 0.4/5 (8.0) | 0.7/5 (14.0) | 1.0/5 (20.0) | 0.7/5 (14.0) | 0.8/5 (16.0) | 1.5/5 (30.0) | 1.8/5 (36.0) | 1.4/5 (28.0) | 0.7/8 (8.8) | 1.3/8 (16.2) | 2.3/8 (28.7) | 1.6/8 (16.2) | 18.9 |
| **CODA-Prompt + KVM** | 7 | **4.5/5 (90.0)** | **4.9/5 (98.0)** | **5.0/5 (100.0)** | **4.8/5 (96.0)** | **4.8/5 (96.0)** | **4.9/5 (98.0)** | **5.0/5 (100.0)** | **4.9/5 (98.0)** | **6.1/8 (76.2)** | **6.6/8 (82.5)** | **7.4/8 (92.5)** | **6.7/8 (83.8)** | **91.1** |

Note: R1/R2/R3 denote the success rate after the 1st/2nd/3rd reset. The format is "completed tasks / total tasks (percentage)". Our method demonstrates exceptional rapid learning, achieving high success rates after just the first reset and near-perfect performance after the second. Other methods show limited learning progress even after three resets.

## H.2 COMPUTATIONAL OVERHEAD ANALYSIS

To quantify the computational overhead introduced by the KVM module, we conducted a detailed complexity analysis on the LIBERO-Long benchmark. The complex, multi-step operations in the LIBERO-Long task sequence provide a realistic testbed for KVM's Smart Gate mechanism.

### H.2.1 KVM INTERVENTION FREQUENCY STATISTICS

Across the 10 long-horizon tasks in LIBERO-Long, we tracked the activation frequency and intervention patterns of the KVM module. As shown in Table 10, KVM's Smart Gate mechanism effectively identifies situations requiring risk assessment, maintaining low-overhead operation in most safe scenarios. "Total Interactions" here refers to every action-step the robot takes with the environment during task execution, including basic operations like grasping, moving, and placing objects.

Table 10: **KVM Intervention Frequency Statistics in LIBERO-Long Tasks.** Statistics are from 1000 complete task sequences (100 runs per task), tracking KVM module activation and intervention.

| Task Type | Total Interactions | Gate Activations | Risk Interventions | Activation Rate (%) | Intervention Rate (%) |
|---|---|---|---|---|---|
| Spatial Navigation | 1,245 | 78 | 12 | 6.3 | 1.0 |
| Object Manipulation | 2,156 | 124 | 28 | 5.8 | 1.3 |
| Goal-oriented Tasks | 1,789 | 98 | 19 | 5.5 | 1.1 |
| Long-horizon Planning | 3,421 | 187 | 41 | 5.5 | 1.2 |
| **Overall** | **8,611** | **487** | **100** | **5.7** | **1.2** |

### H.2.2 QUANTITATIVE ANALYSIS OF COMPUTATIONAL COMPLEXITY

We quantified the computational overhead of the KVM module across several dimensions, as shown in Table 11. Compared to the baseline CODA-Prompt model, the additional computational cost introduced by KVM is acceptable, while significantly improving system safety.

Table 11: **Detailed Analysis of KVM Module Computational Overhead.** Metrics measured on an NVIDIA A100-80GB GPU.

| Metric | CODA-Prompt | CODA-Prompt + KVM | Increase | Relative Increase |
|---|---|---|---|---|
| Inference Latency (ms/step) | 12.4 | 13.7 | +1.3 | +10.5% |
| Memory Usage (GB) | 4.2 | 4.8 | +0.6 | +14.3% |
| GPU Utilization (%) | 78.5 | 82.1 | +3.6 | +4.6% |
| *Under conditional activation:* | | | | |
| Context Encoding Only (ms) | - | 0.8 | +0.8 | - |
| Full Risk Assessment (ms) | - | 2.1 | +2.1 | - |
| Energy Function Calc. (ms) | - | 1.2 | +1.2 | - |
| *In safety-critical scenarios:* | | | | |
| Risk Avoidance Success (%) | 17.2 | 91.1 | +73.9 | +430% |
| Catastrophic Failures Reduced (%) | - | - | -87.3 | -87.3% |

**Key Findings:**

(1) **Low-Frequency, High-Efficiency Intervention:** KVM's Smart Gate activates risk assessment in only 5.7

(2) **Acceptable Performance Overhead:** KVM adds only a 10.5

(3) **Significant Safety Gains:** In safety-critical scenarios, KVM boosts risk avoidance success from 17.2

(4) **Adaptive Computation Strategy:** In 95

