# OpenReview forum: "Contextual Forgetting: Mitigating Knowledge Obsolescence for Safe Lifelong Robot Learning"
_ICLR.cc/2026/Conference — Submitted to ICLR 2026_

### Official Review · Reviewer_a1oh · 2025-10-26

**Soundness:** 2
**Presentation:** 2
**Contribution:** 2
**Rating:** 4
**Confidence:** 4

**Summary:**

The authors propose a contextual forgetting algorithm to enable autonomous agents to forget knowledge they have once learned but is inappropriate in new contexts, e.g., after a distribution shift. The key component is a knowledge validity module that maps contexts into a Gaussian distribution, whose mean and variance may activate a dynamic safety threshold. Risk is then assessed via an energy-based model. Experiments on different robot benchmarks show that that the novel method outperforms baseline methods.

**Strengths:**

1) Continual or lifelong learning is a relevant topic, and forgetting plays a central role. As such, the contribution is timely.

2) The evaluation and comparison to the state-of-the-art is extensive.

**Weaknesses:**

1) I don't quite follow the central claim that there is no mechanism for "how to forget." When typing something like "learning to forget" in Google Scholar, various papers pop up that have developed such mechanisms. Also, context-based learning has been largely explored. Also there, old knowledge is not applied anymore as soon as we are in a new context. I think there is some more discussion of related literature required to adequately position the contributions of the paper.

2) The authors promise a detailed theoretical justification in the appendix. I think it would be good to add some key elements to the main body. Also, the theorem in the appendix is missing a proof.

3) The description of the method lacks important details. What exactly is the context $c_\mathrm{raw}$? Is this something we are somehow observing? Or that we need to learn? How does $\tau_\mathrm{adaptive}$ enter the algorithm exactly? What is the feature representation $\mathbf{h}$? What is $\alpha_t$ that is mentioned in Section 3.2?

4) The appendix seems a bit disorganized with various margin violations.

**Questions:**

1) Can you clarify how your contributions relate to existing "learning to forget" frameworks and to general context-based learning?

2) Is $c_\mathrm{raw}$ something that we somehow assume as given, or is this something that is learned from observations?

3) Can you clarify the algorithmic details (How does $\tau_\mathrm{adaptive}$ enter the algorithm exactly? What is the feature representation $\mathbf{h}$? What is $\alpha_t$ that is mentioned in Section 3.2?)?

4) Can you provide some motivation/justification for the form of the validity score?

5) What exactly is the feature representation $\mathbf{h}$ that is used in the computation of the energy function?

---

### Official Review · Reviewer_pxpu · 2025-10-28

**Soundness:** 2
**Presentation:** 2
**Contribution:** 2
**Rating:** 4
**Confidence:** 3

**Summary:**

This paper aims to address the “knowledge obsolescence” issue in continual learning, which means that a skill, once optimal in a specific context, might become a direct cause of catastrophic failure. To address this issue, the paper proposes Knowledge Validity Module (KVM) to actively identify and mitigate hazardous interactions caused by knowledge inapplicability. By integrating KVM with a continual learning algorithm CODA-Prompt, the paper conducted experiments to demonstrate that KVM significantly reduces catastrophic failures caused by knowledge obsolescence without sacrificing learning efficiency.

**Strengths:**

1.	The studied “knowledge obsolescence” problem is interesting and underexplored in the literature.

2.	The experimental results shown in the paper are good.

**Weaknesses:**

1.	The presentation can be further improved. First, some important preliminary information is not provided, such as the basic setting of this work (and CODA-Prompt) , which makes it difficult to understand and evaluate the method section. Second, in section 3, the KVM is naively introduced without sufficient discussion. For example, why does KVM map raw multimodal context into a Gaussian distribution? What is the CL core? Third, in the experimental part, the paper makes some claims without solid empirical support. For example, in line 302, the paper says, “We attribute this to KVM’s ”Contextual Forgetting” mechanism”. Since the proposed method is CODA-Prompt + KVM and CODA-Prompt is missing in table 2, it’s not appropriate to claim that the benefit is from KVM not CODA-Prompt. And the discussion from line 370 to line 374 also looks subjective.

2.	The “knowledge obsolescence” is an interesting topic. But the paper use examples like first “lifting hard pizza” task and then “lifting soft pizza” to highlight the “knowledge obsolescence”, which is not convincing. “lifting hard pizza” task and “lifting soft pizza” seem like two different tasks, and the failure of “lifting soft pizza” might be because this is a new task the model needs to learn but not “knowledge obsolescence”. In language models, an example of “knowledge obsolescence” is that the current President of USA is Trump not Biden who was President two years ago. Some other works, like knowledge/representation editing[1,2], also study “knowledge obsolescence” in language models.

3.	It's claimed that the proposed method may significantly reduce catastrophic failures caused by knowledge obsolescence without sacrificing learning efficiency. I'm curious whether the KVM has a negative effect on previous capability, such as the performance of “lifting hard pizza” after learning “lifting soft pizza”?


[1] Mass-Editing Memory in a Transformer
[2] EasyEdit: An Easy-to-use Knowledge Editing Framework for Large Language Models

**Questions:**

1.	How to choose the hyperparameter k?

2.	Some format issues in subtitle 4.3, page 31, page 34.

3.	What is the definition of catastrophic failures? What is the Catastrophic Failure Rate in Table 4?

---

### Official Review · Reviewer_bUwU · 2025-11-01

**Soundness:** 2
**Presentation:** 1
**Contribution:** 3
**Rating:** 2
**Confidence:** 3

**Summary:**

This paper addresses the problem of knowledge obsolescence in lifelong robot learning, where once useful knowledge becomes unsafe or harmful as the environment changes. Rather than focusing on preserving knowledge, the authors discuss mechanisms to discard outdated or unsafe information (forgetting) intentionally.

Contributions:

1. The paper introduces a mechanism, "Contextual Forgetting" via a KVM that evaluates the relevance and safety of learned policies over time.

2. Results indicate that CF–KVM reduces catastrophic failures caused by outdated knowledge while maintaining comparable learning efficiency to baseline continual learning methods.

**Strengths:**

1. The paper challenges the conventional assumption in continual learning that forgetting is always harmful. The perspective on how learning systems might handle outdated or risky behaviors is interesting.

2. The topic of knowledge obsolescence and its link to safety is timely.

**Weaknesses:**

Major concerns:

1. The paper emphasizes safety and hazard reduction. However, all experiments are conducted in simulation using LIBERO and SimplerEnv, which are generalization and robustness benchmarks rather than safety evaluations. There are no quantitative safety metrics, no modeling of physical risk, and no real-robot validation.

2. The contribution of the proposed Knowledge Validity Module (KVM) and its Energy-Based Model component is unclear. The ablation study in Table 4 reports small numerical differences but lacks statistics such as standard deviation, mechanistic interpretation, and a clear definition of "catastrophic failure".

3. The paper uses an energy defined in Eq. 5-6. However, such an energy framework cannot distinguish novel-but-safe from old-and-dangerous interactions, as both are far from past safe examples in feature space. This causes either: (1) false negatives allowing catastrophic old policies, or (2) false positives blocking necessary safe exploration. The method section should include relevant discussions.

**Questions:**

Q1: Can the authors provide more discussions and justifications on real-robot validation with quantitative physical risk metrics?

Q2: Table 4 shows only 1.9% improvement (70.5% vs 68.6%) when adding KVM. Is this statistically significant?

Q3: How does the energy model (Eq. 5-6) distinguish novel-but-safe from old-and-dangerous interactions? Can authors provide precision/recall metrics for the experiment?

---

### Meta-Review · Area_Chair_jgk3 · 2025-12-21

**Summary:**

The authors propose a contextual forgetting algorithm to enable autonomous agents to forget knowledge they have once learned but is inappropriate in new contexts. Essentially it is a continual learning algorithm that prevents catastrophic failures in general performance but ensures that you do forget the targeted harmful task. All reviewers mention that to some degree, the experiments seem to validate it. However, all reviewers also agree that definitions for important terms are missing, the benchmarks as well as metrics are not exactly appropriate, and various handwavy phrases and justifications are used throughout the paper. It would have been helpful to have a rebuttal from the authors to argue their perspective on it. However, absent a rebuttal, I will not overturn the reviewer's comments and all reviewers unanimously agree that the paper is not ready for acceptance. Thus, I recommend rejection.

**Reviewer Concerns:**

No rebuttal.

**Reviewer Scores:**

None because authors did not respond.

---

### Decision · Program_Chairs · 2026-01-26

Reject